# An external sodium ion binding site controls allosteric gating in TRPV1 channels

**Andres Jara-Oseguera, Chanhyung Bae, Kenton J Swartz\***

Molecular Physiology and Biophysics Section, Porter Neuroscience Research Center, National Institute of Neurological Disorders and Stroke, National Institutes of Health, Bethesda, United States

**Abstract** TRPV1 channels in sensory neurons are integrators of painful stimuli and heat, yet how they integrate diverse stimuli and sense temperature remains elusive. Here, we show that external sodium ions stabilize the TRPV1 channel in a closed state, such that removing the external ion leads to channel activation. In studying the underlying mechanism, we find that the temperature sensors in TRPV1 activate in two steps to favor opening, and that the binding of sodium to an extracellular site exerts allosteric control over temperature-sensor activation and opening of the pore. The binding of a tarantula toxin to the external pore also exerts control over temperature-sensor activation, whereas binding of vanilloids influences temperature-sensitivity by largely affecting the open/closed equilibrium. Our results reveal a fundamental role of the external pore in the allosteric control of TRPV1 channel gating and provide essential constraints for understanding how these channels can be tuned by diverse stimuli.

**\*For correspondence:** swartzk@ninds.nih.gov

## Introduction

Transient receptor potential (TRP) channels belong to a large and diverse superfamily of tetrameric cation channels that contain six transmembrane helices per subunit (*Liao et al., 2013*; *Hellmich and Gaudet, 2014*), similar to voltage-activated ion channels (*Long et al., 2007*), ryanodine receptors (*Yan et al., 2015*) and IP$_3$ receptors (*Fan et al., 2015*) (*Figure 1A*). Several TRP channels are exquisitely temperature-dependent, enabling them to serve as biological thermosensors throughout the animal kingdom (*Vay et al., 2012*). The vanilloid-sensitive TRPV1 channel is a weakly voltage-dependent and cation-selective channel that is expressed in nociceptive sensory neurons, where it acts as a sensor for noxious stimuli (*Caterina and Julius, 2001*) including heat, extracellular acidic pH, vanilloid compounds (e.g capsaicin), and venom toxins (e.g. double-knot toxin or DkTx) (see [*Hilton et al., 2015*] for review). The sensitivity of TRPV1 to diverse modulators is fascinating because many stimuli are capable of influencing the responses of TRPV1 to other signals. For example, protons can sensitize the channel to activation by heat (*Jordt et al., 2000*) and capsaicin (*Jordt et al., 2000*; *Ryu et al., 2003*), and both capsaicin (*Voets et al., 2004*; *Matta and Ahern, 2007*; *Gregorio-Teruel et al., 2014*) and protons (*Lee and Zheng, 2015*) can sensitize the channel to activation by voltage. Conceptually, these observations can be explained by allosteric models wherein distinct stimulus-sensing domains are coupled to each other (*Latorre et al., 2007*), and to an internal gate within the pore that opens and closes the ion permeation pathway (*Salazar et al., 2009*; *Cao et al., 2013*) (*Figure 1A*).

The regions of TRPV1 targeted by some activating stimuli have been identified, including the vanilloid binding site within internal regions of the S1-S4 domain (*Jordt and Julius, 2002*; *Cao et al., 2013*), and both proton- (*Jordt et al., 2000*) and DkTx-binding sites within the external

**eLife digest** Humans and other mammals sense elevated heat and other painful stimuli via a sensory ion channel protein called TRPV1. Ion channels create pores in the outer membrane of cells and act as gates that open and close to regulate the flow of ions into and out of cells. This flow of ions generates electrical signals that sensory neurons use to communicate information about the environment to the brain. The TRPV1 channel is opened by heat, but also by venom toxins, acid and the active ingredient in hot chilli peppers.

The fluid that surrounds animal cells often contains high levels of sodium ions, and these ions flow through TRPV1 channels when they are open. Also, when sodium ions are removed from the fluid surrounding a cell, TRPV1 channels will spontaneously open. It is not known why this happens, but it suggests that sodium ions help to regulate the activity of the TRPV1 channel, and despite having been extensively studied, it remains unclear how this channel senses temperature and responds to other harmful stimuli.

Jara-Oseguera et al. have now investigated the role of sodium ions in activating TRPV1 in cells grown in the laboratory. The experiments involved a technique that records the movement of ions through TRPV1 channels in the cell surface membrane while exposing the cell to various stimuli that open or close the channels. The results showed that TRPV1 contains an important site on its outer surface that binds sodium ions and fine-tunes the channel's properties. This enables the channel to be activated by potentially damaging temperatures. Further analysis revealed that TRPV1 undergoes at least two distinct physical changes in response to heat and that the binding of sodium ions to the channel's outer structure regulates one of these changes.

Jara-Oseguera et al. went on to discover that the binding of a tarantula toxin to the outer surface of TRPV1 renders the channel insensitive to changes in temperature. These findings suggest that structural changes to the external part of this channel protein are closely associated with its ability to sense temperature, possibly because this region of the protein is directly responsible for detecting changes in temperature.

These findings shed light on how this channel detects and responds to dangerously high temperatures, as well to other harmful stimuli. An aim for future studies is to pinpoint the specific regions of the protein that form the sodium-binding sites and to test the hypothesis that temperature sensors are located on the outer surface of the protein.

pore (*Bohlen et al., 2010*; *Cao et al., 2013*; *Bae et al., 2016*) (*Figure 1A*). However, we currently do not understand the structural basis by which the binding of protons, capsaicin or DkTx promotes opening of the internal pore. Moreover, there is no consensus about the site and mechanism of temperature sensitivity. Mutations throughout TRPV1 alter the temperature sensitivity of the channel (*Feng, 2014*), and distinct models have been proposed to explain temperature sensitivity in these channels (*Voets, 2012*). For example, in the conventional allosteric model, a specialized temperature-sensitive transition is coupled to a temperature- and voltage-insensitive opening transition (*Brauchi et al., 2004*; *Matta and Ahern, 2007*). In contrast, others have suggested that temperature sensitivity arises from a weakly voltage-sensitive opening transition (*Voets et al., 2004*). Indeed, one recent study on voltage-activated potassium (Kv) channels suggests that the opening transition in these related channels can be made steeply temperature-dependent when the coupling between the voltage-sensing and pore domains is weakened (*Yang et al., 2014*). It has also been suggested that TRP channels may not contain localized temperature-sensors, but that their steep temperature-dependence results from changes in solvation of hydrophobic residues throughout the protein (*Clapham and Miller, 2011*).

Recent landmark near-atomic resolution structures of the TRPV1 channel confirm the presence of an internal gate related to that present in Kv channels (*Liu et al., 1997*; *del Camino and Yellen, 2001*). In addition, the external pore and selectivity filter change conformation in the combined presence of DkTx and the vanilloid resiniferatoxin (RTx) (*Cao et al., 2013*; *Liao et al., 2013*). Given our limited understanding of the temperature-sensitivity of TRPV1, it is unclear whether these structures provide insight into the mechanisms of heat activation. However, the structural rearrangements

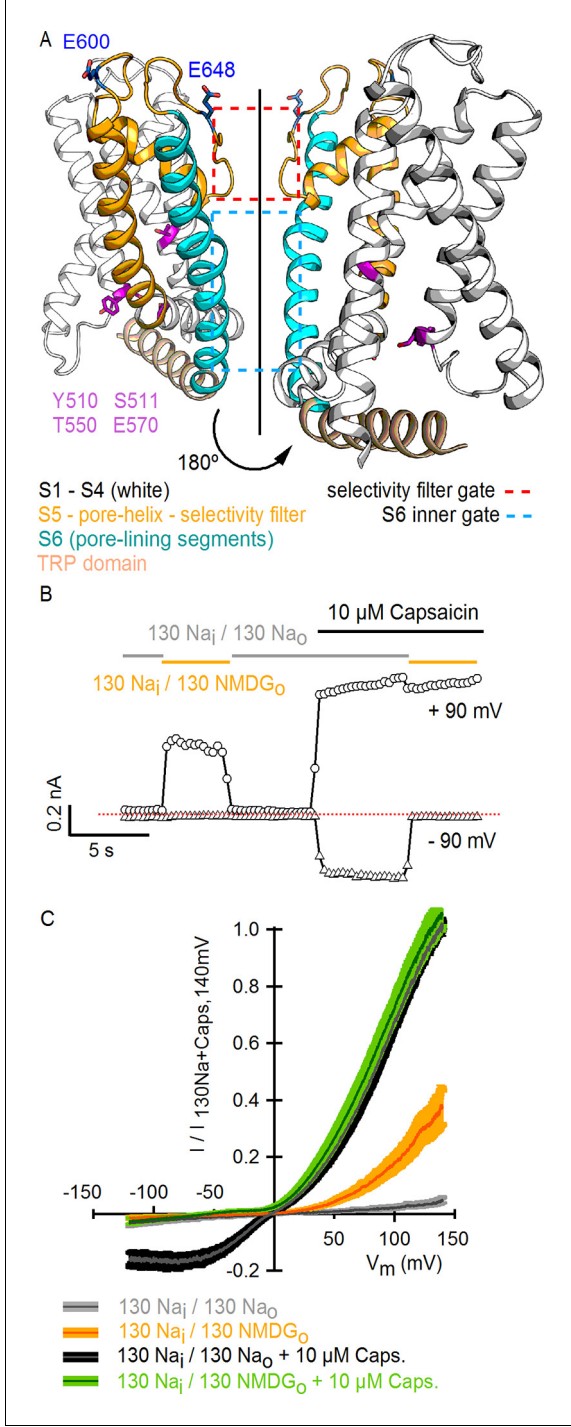

**Figure 1.** Substitution of extracellular $Na^+$ with $NMDG^+$ increases TRPV1-mediated currents. (**A**) Side view in ribbon representation of the transmembrane domains of two opposing TRPV1 subunits (as indicated by the black arrow, extracellular face on the top, intracellular face on the bottom) in the apo state (refined TRPV1 structural model [**Bae et al., 2016**]). The dashed boxes denote the location of the two constrictions proposed to serve as gates. Side chains of residues forming the binding site for capsaicin (purple) or determining activation of TRPV1 by protons (blue) are shown as sticks. (**B**) Representative time-course of whole-cell TRPV1-mediated currents elicited by 100-ms voltage pulses from -90 mV (triangles) to +90 mV (circles) at 300 ms intervals and at room temperature. The colored horizontal lines signal the onset of rapid-solution exchange as indicated by the labels. The dotted red line indicates the zero-current level. (**C**) Normalized TRPV1 current-voltage (I-V) relations obtained from 1s-duration

*Figure 1 continued on next page*

*Figure 1 continued*

voltage-ramps, following the same solution-exchange sequence as in (**B**). The darker curves are the mean and lighter-colored envelopes the standard error (n = 8).

The following figure supplements are available for figure 1:

**Figure supplement 1.** Substitution of external Na$^+$ with NMDG$^+$ induces channel rundown at room temperature with high cell-to-cell variability.

**Figure supplement 2.** Activation of rat TRPV1 channel orthologues by substituting external Na$^+$ with NMDG$^+$.

**Figure supplement 3.** Comparison of TRPV1 channel I-V relations measured using voltage steps and voltage ramps.

**Figure supplement 4.** Theoretical I-V relations in the presence and absence of external Na$^+$ obtained with the Goldman-Hodgkin-Katz current equation.

within the external pore are intriguing because mutations in this region have been reported to alter activation of the channel by protons (*Jordt et al., 2000*) and heat (*Grandl et al., 2010*; *Cui et al., 2012*; *Yang et al., 2014*). In addition, external Na$^+$ ions have been reported to inhibit the TRPV1 channel at room temperature (*Ohta et al., 2008*), raising the possibility that an external cation binding site may regulate these channels. In the present study, we demonstrate the presence of a critical external Na$^+$ ion-binding site that must be occupied for the channel to remain closed at room temperature, and to be available for opening by noxious heat. Our results also show that multiple steeply temperature-dependent transitions are allosterically coupled to an opening transition, and that binding of Na$^+$ or DkTx to the external pore strongly influences the activity of the temperature-sensor and opening of the pore.

## Results

To investigate the influence of external Na$^+$ on TRPV1 channels, we obtained whole-cell patch clamp recordings of cells expressing rat TRPV1 using internal and external solutions containing Na$^+$ (130 mM) as the sole charge carrier, and then exchanged the external Na$^+$ solution with one containing N-methyl-D-glucamine (NMDG$^+$; 130 mM) (*Figure 1B*). At room temperature, this simple manipulation produced a rapid and reversible increase in outward currents at positive voltages (*Figure 1B*), consistent with previous results (*Ohta et al., 2008*). Subsequent application of a saturating concentration of capsaicin further increased the outward currents independently of whether external Na$^+$ or NMDG$^+$ was present (*Figure 1B*). At moderate TRPV1 expression levels, where outward Na$^+$ currents could be accurately measured, the inward currents with external NMDG$^+$ were negligible even in the presence of capsaicin (*Figure 1B*), indicating that TRPV1 has a relatively low permeability to large cations. Although large outward currents induced by exchanging external ions were observed in all cells in which capsaicin evoked measurable currents, this manipulation also produced a variable and irreversible rundown of the channel that increased with time in Na$^+$-free solutions (*Figure 1—figure supplement 1*). Exchanging external Na$^+$ with NMDG$^+$ also produced large outward currents in human, mouse and chicken TRPV1 orthologues (*Figure 1—figure supplement 2*), which together with the previous report on porcine TRPV1 (*Ohta et al., 2008*), suggest that this phenomenon is a conserved feature of TRPV1 channels. Given the robust effects of exchanging external Na$^+$ with NMDG$^+$, we set out to explore the underlying mechanism.

### An external Na$^+$ site stabilizes a closed state of TRPV1

To investigate the extent to which the large outward currents in external NMDG$^+$ result from a change in driving force when exchanging external solutions, we obtained current-voltage (I-V) relations using rapid voltage ramps (*Figure 1C*) or voltage steps in cells exhibiting minimal rundown at room temperature (*Figure 1—figure supplement 3*). I-V relations for fully activated channels (10 μM capsaicin) in the presence of external Na$^+$ or NMDG$^+$ superimpose at positive voltages (*Figure 1C*

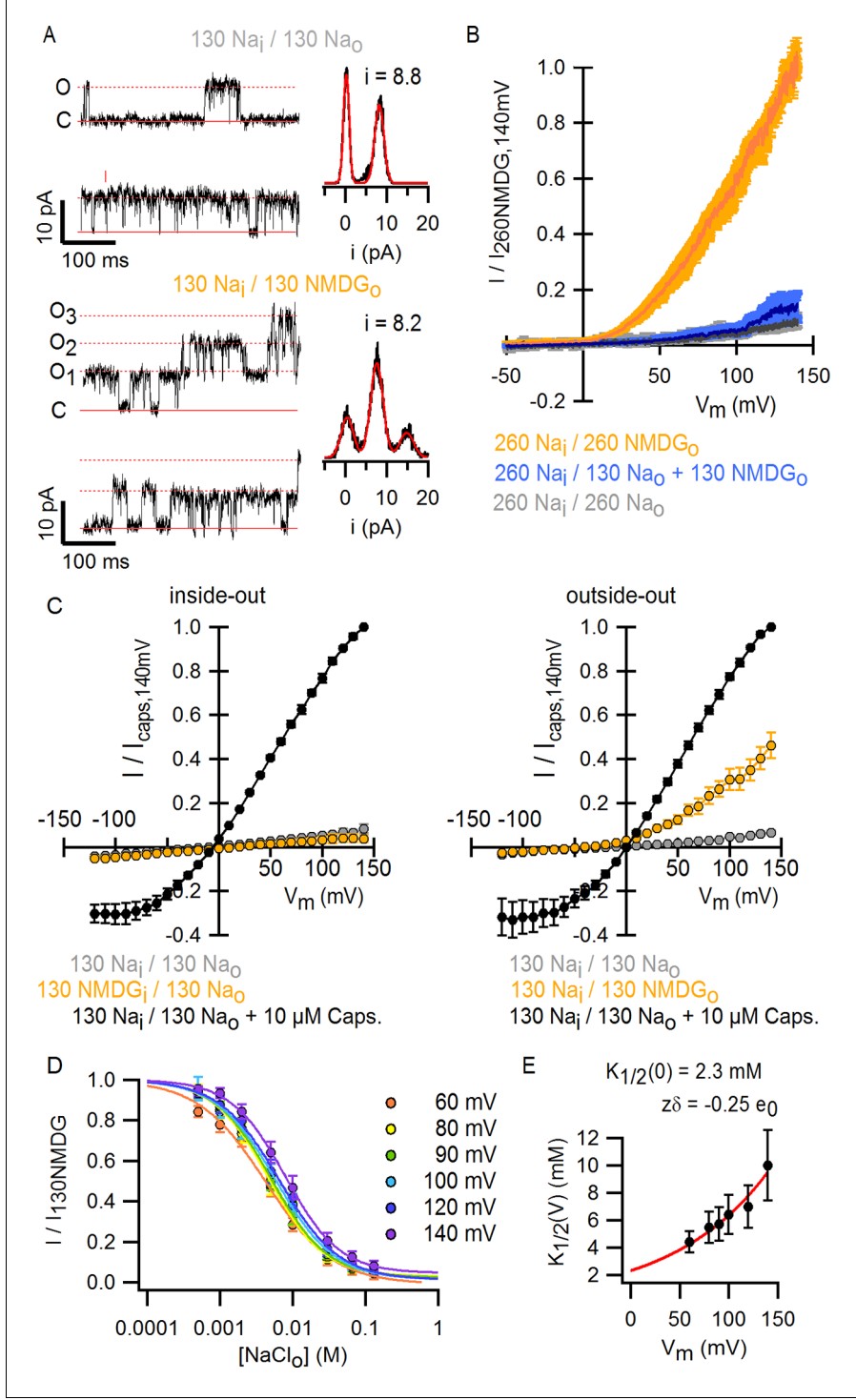

**Figure 2.** Extracellular sodium ions are allosteric inhibitors of the TRPV1 channel. (**A**, left) Representative recordings at +90 mV performed on outside-out patches containing a few TRPV1 channels in the presence of 130 mM external Na+ (top) or NMDG+ (bottom) at room temperature. The continuous horizontal line represents the zero-current level (C, all channels closed) and the dotted red lines indicate current value levels corresponding to one ($O_1$), two ($O_2$) or three ($O_3$) simultaneously open channels. (**A**, right) All-points current amplitude histograms constructed from the recordings on the left. The red curves are fits to a sum of Gaussian functions. (**B**) Normalized I-V relations constructed from voltage-ramps measured in the whole-cell configuration with 260 mM internal Na+ ($Na_i$) and the extracellular solutions indicated in the figure. The dark curves are the mean and the lighter-colored envelopes the standard error (n = 4). (**C**) I-V relations obtained from families of 100 ms voltage-pulses in either the

*Figure 2 continued on next page*

*Figure 2 continued*

inside-out (left) or outside-out (right) configuration (mean ± SEM, n = 4–5). (**D**) Sodium dose-response relations (mean ± SEM, n = 7) obtained from voltage-ramps in the whole-cell configuration using solutions with different $Na^+/NMDG^+$ ratios, all adding up to a total cation concentration of 130 mM. The continuous curves are fits to the Hill equation with a Hill coefficient of 1.2 ± 0.1. (**E**) $K_{1/2}$-V relation from fits as in (**D**). The red curve is a fit to $K_{1/2}(V)$ = $K_{1/2}(0)$ x exp(-$z\delta V/k_BT$).

The following figure supplement is available for figure 2:

**Figure supplement 1.** Monovalent cation selectivity of the external $Na^+$-binding site in TRPV1.

and *Figure 1—figure supplement 3*), suggesting that the unitary conductance of TRPV1 channels is similar under these conditions. In addition, theoretical I-V relations calculated from the Goldman-Hodgkin-Katz equation predict little change in unitary conductance at positive voltages (*Figure 1— figure supplement 4*), and we confirmed that exchanging external solutions does not affect the single-channel current amplitude (i) at positive voltages in outside-out patches containing a few channels (*Figure 2A*). From these results, we conclude that exchanging external $Na^+$ with $NMDG^+$ leads to spontaneous opening of the TRPV1 channel at room temperature.

To distinguish whether channel activation results from the removal of $Na^+$ or the addition of $NMDG^+$, we obtained voltage ramps in 130 mM $NMDG_o$ in the additional presence of 130 mM $Na_o$ (with 260 mM internal $Na^+$) and observed only small outward currents at positive voltages that superimposed with those obtained in the presence 260 mM $Na_o$ (*Figure 2B*). This result indicates that $NMDG^+$ does not activate the channel in the presence of $Na^+$, but rather removal of external $Na^+$ leads to channel activation. Removal of internal $Na^+$ in recordings using inside-out patches produced no channel activation (*Figure 2C*, left panel), whereas removing external $Na^+$ using outside-out patches led to robust activation (*Figure 2C*, right panel), demonstrating that the inhibitory influence of $Na^+$ is limited to the external side of the TRPV1 channel. We also measured the concentration-dependence for external $Na^+$ inhibition over a range of voltages (*Figure 2D*) and found that inhibition was only weakly voltage-dependent, with an extrapolated apparent affinity ($K_{1/2}$) at 0 mV of 2 mM (*Figure 2E*). These results suggest that external $Na^+$ stabilizes a closed state of the TRPV1 channel by binding to a site on the extracellular side of the protein. This ion-binding site is not highly selective for $Na^+$, as external $K^+$ can stabilize the closed state almost as effectively as $Na^+$, followed by $Rb^+$, $Tris^+$, $Cs^+$ and $Li^+$ (*Figure 2—figure supplement 1*).

## External $Na^+$, protons and DkTx modulate the TRPV1 through overlapping mechanisms

An external $Na^+$ binding site that stabilizes TRPV1 in a closed state is interesting because titratable acidic residues are commonly found in such sites (*Lev et al., 2013*) and acid is a physiological stimulus for this channel, with mildly acidic conditions sensitizing the channel to either capsaicin or heat, and strongly acidic conditions causing channel activation (*Jordt et al., 2000;Ryu et al., 2003*). To explore a possible mechanistic link between the external $Na^+$ site and acid regulation, we changed external pH from 7.4 to 6 and observed a pronounced shift in the concentration-dependence for $Na^+$ inhibition to higher $Na^+$ concentrations (*Figure 3A*). Moreover, activation of the channel by pH 6 could be completely overcome by raising the $Na^+$ concentration to 300 mM. These results are consistent with the expectation that protonation of acidic residues forming the $Na^+$ binding site would lower the affinity of the metal ion, suggesting that the inhibitory $Na^+$ binding site is intimately involved in the regulation of the TRPV1 channel by acid. However, acidic solutions further stimulated channel activation in the absence of external $Na^+$ (*Figure 3A*), indicating that protons also activate TRPV1 channels through a $Na^+$-independent mechanism. These findings support the idea that proton-mediated channel activation and proton-dependent potentiation of responses to other stimuli have distinct mechanisms (*Jordt et al., 2000*; *Ryu et al., 2007*).

We next set out to determine whether other stimuli acting on the extracellular pore activate TRPV1 through a mechanism associated with the binding of external $Na^+$. The double-knot tarantula toxin (DkTx) is an interesting candidate as it activates the TRPV1 channel with high avidity by binding to the periphery of outer pore of the channel (*Figure 3—figure supplement 1A*) (*Bohlen et al.,*

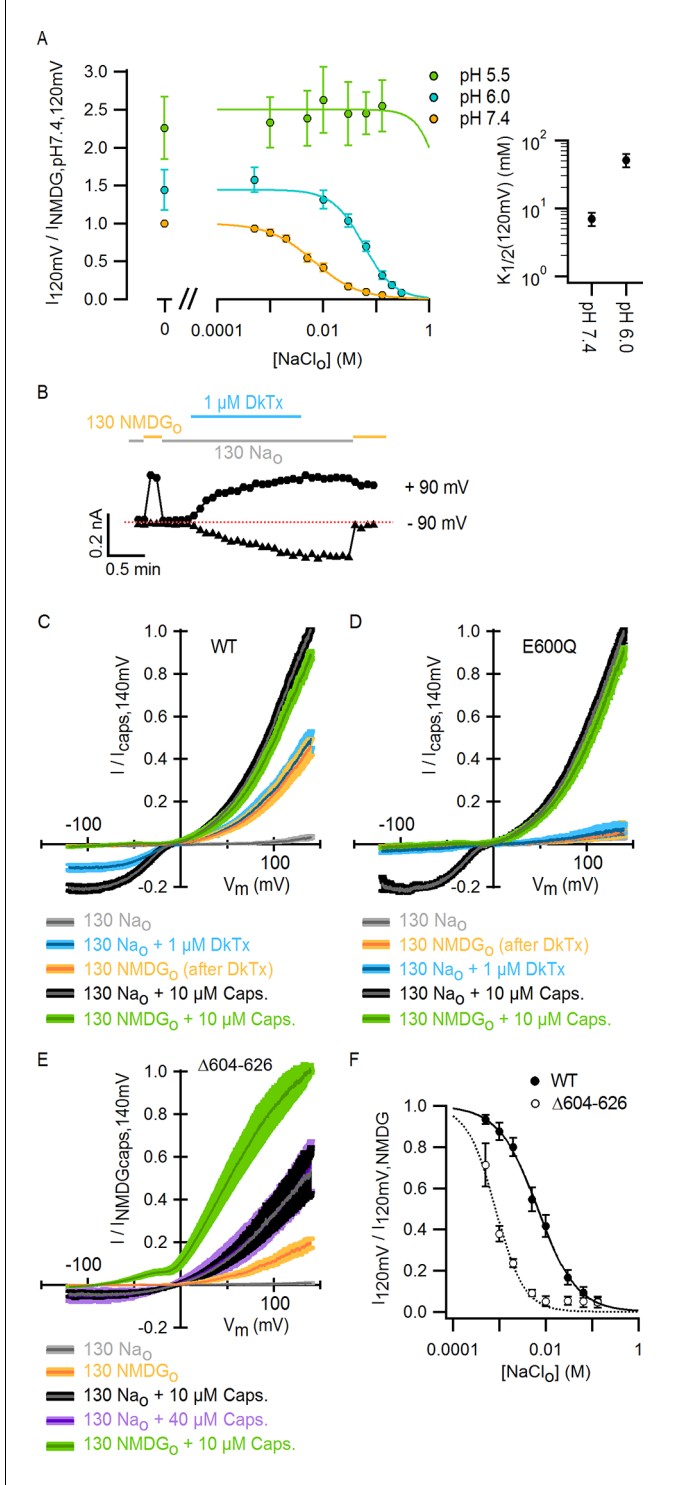

**Figure 3.** External Na⁺, H⁺ and DkTx modulate the TRPV1 channel through overlapping mechanisms involving E600 in the extracellular pore. (**A**) Extracellular Na⁺ dose-response relations measured at different extracellular pH values at +120 mV in the whole-cell configuration in response to voltage ramps (mean ± SEM, n = 3–7). Experiments at different pH values were recorded from independent cells at room temperature. The data on the left of the axis break reflect activation of TRPV1 channels by protons in the absence of external Na⁺. The continuous curves are fits to the Hill equation. The obtained parameters were: s = 1.2 ± 0.08 and $K_{1/2}$ = 7 ± 0.15 mM at pH 7.4 (n = 7); s = 1.5 ± 0.08 and $K_{1/2}$ = 51.2 ± 11.5 mM at pH 6.0 (n = 3). The Hill equation was not fit to the data at pH 5.5 (n = 5), but the Hill function shown (green curve) has s = 2.0 and $K_{1/2}$ = 2.0 M. Mean $K_{1/2}$ values

*Figure 3 continued on next page*

*Figure 3 continued*

from the fits to data at pH 7.4 and 6.0 are shown on the right panel insert. For all cells, data were normalized to the currents measured in the absence of external $Na^+$ at pH 7.4 (130 mM $NMDG^+$ for pH 7.4 and 5.5; 300 mM $NMDG^+$ for pH 6.0). (B) Representative whole-cell WT TRPV1 current time-course at -90 and +90 mV constructed from a train of voltage-ramps at room temperature. Horizontal thick lines denote the removal of external $Na^+$ (yellow) or the application of DkTx (blue). The dotted line denotes the zero-current level. (C) Mean normalized I-V relations (mean – thin darker curves – ± SEM – lighter-colored envelopes, n = 4) from WT TRPV1 channels obtained from whole-cell recordings at room temperature in response to voltage ramps with 130 mM intracellular $Na^+$ and the extracellular solutions indicated by the labels at the bottom. The order of the labels reflects the order in which the different solutions were tested in each experiment. (D) Mean normalized I-V relations (mean ± SEM, n = 4) from TRPV1 channels with the E600Q mutation, obtained as in (C). In these experiments, the 0 $Na_o$-solution was tested before application of DkTx. (E) Normalized I-V relations for TRPV1 Δ604–626 obtained in the whole-cell configuration at room temperature (continuous curves are the mean, lighter envelopes the SEM, n = 6) in response to voltage ramps. (F) External $Na^+$ dose-response relations for TRPV1 Δ604–626 (open circles, mean ± SEM, n = 4) measured at +120 mV from voltage-ramps in the whole-cell configuration. The dotted curve is a fit to the Hill equation with parameters: s = 2.2 ± 0.7, $K_{1/2}$ = 0.7 ± 0.1 mM. Closed circles are data for WT TRPV1 at +120 mV (*Figure 2D* and *3A* - pH 7.4). The continuous curve is a fit to the Hill equation with parameters indicated in (A).

The following figure supplement is available for figure 3:

**Figure supplement 1.** The location of E600, the extracellular pore turret and the binding site for DkTx within the outer pore of TRPV1, and the role of external $Mg^{2+}$ ions in TRPV1 modulation.

*2010*; *Cao et al., 2013*; *Bae et al., 2016*). We recently solved the solution structure of DkTx and performed a detailed analysis of its interactions with the TRPV1 channel, which suggested that the toxin activates TRPV1 by disrupting a cluster of hydrophobic residues at the extracellular half of the S5 and S6 segments (*Bae et al., 2016*). Interestingly, removal of external $Na^+$ before application of the toxin activates TRPV1 to a similar extent when compared with a saturating concentration of DkTx (*Figure 3B and C*), and removal of external $Na^+$ did not produce further activation of DkTx-bound channels. In contrast, application of a saturating concentration of capsaicin produced further activation (*Figure 3C*). The lack of additivity for the activation of TRPV1 by DkTx and the removal of external $Na^+$ contrasts with our results with acidic extracellular pH in the absence of external $Na^+$ (*Figure 3A*), and suggests that DkTx and external $Na^+$ modulate the TRPV1 through convergent mechanisms.

The unambiguous identification of the $Na^+$-binding site would require a high resolution structure of TRPV1. However, a glutamate at position 600 near the top of S5 (*Figure 3—figure supplement 1A*) has been previously associated with the inhibition of TRPV1 by external $Na^+$ (*Ohta et al., 2008*), and has also been shown to be involved in the proton-dependent potentiation of TRPV1 responses to heat and capsaicin (*Jordt et al., 2000*). To confirm a key role of this residue in modulation of TRPV1 by external $Na^+$, we investigated the effect of removing external $Na^+$ on E600Q mutant channels, and found that the removal of external $Na^+$ no longer produced constitutive activation even though the mutant remained responsive to capsaicin (*Figure 3D*). Consistent with DkTx and external $Na^+$ having convergent mechanisms for modulating TRPV1, E600Q channels were not responsive to a concentration of DkTx that is saturating in WT channels (*Figure 3D*). These results demonstrate an important role for E600 in mediating channel inhibition by external $Na^+$ and activation by DkTx, and suggest that this residue may be a part of the $Na^+$ binding site.

Another interesting region within the outer pore of TRPV1 is the pore turret (Δ604–626, *Figure 3—figure supplement 1B*), a stretch of 22 residues that was removed in the construct of TRPV1 recently used for structure determination (*Liao et al., 2013*) and that is located near to E600 and to the binding site for DkTx (*Figure 3—figure supplement 1A*). We therefore explored whether the deletion of the pore turret retains modulation by external $Na^+$. Initially, we found that the removal of external $Na^+$ produced robust channel activation at depolarized potentials in channels lacking the pore turret (*Figure 3E*), and that the addition of 10 μM capsaicin in the presence of external $Na^+$ further activated the channels, similar to WT TRPV1. Interestingly, the removal of the pore turret increased the apparent affinity of the channel for $Na^+$ (*Figure 3F*), indicating that the pore turret does not form the $Na^+$ binding site, but influences the cation-binding site through an

allosteric mechanism. However, we also made the surprising discovery that the removal of external Na$^+$ in the presence of 10 μM capsaicin resulted in enhanced channel activation at positive voltages relative to 10 μM capsaicin in the presence of 130 mM external Na$^+$ (*Figure 3E*), in stark contrast with WT channels where I-V relations under these two conditions superimpose at depolarized potentials due to maximal channel activation (*Figure 1C* and *Figure 3C*). Increasing the concentration of capsaicin from 10 to 40 μM did not further activate Δ604–626 TRPV1 channels in the presence of external Na$^+$ (*Figure 3E*), indicating that 10 μM is a saturating concentration and that capsaicin is a partial agonist for this construct. This reveals that the pore turret is mechanistically associated with the modulation of the channel not only by stimuli acting on the extracellular pore (e.g. external Na$^+$), but also by capsaicin that binds near the intracellular part of the protein (*Figure 1A*).

Finally, we were interested in determining whether external Mg$^{2+}$ ions activate the TRPV1 channel directly, as was suggested in two previous studies that analyzed the effect of substituting external Na$^+$ with Mg$^{2+}$ without considering the inhibitory effects of external Na$^+$ (*Cao et al., 2014*; *Yang et al., 2014*). We tested the ability of 130 mM external MgCl$_2$ to activate the TRPV1 channel in the presence of 130 mM external NaCl (and 260 mM intracellular NaCl), as we did previously in *Figure 2B* to determine whether NMDG$^+$ was an activator of TRPV1. Contrary to what we observed for NMDG$^+$, Mg$^{2+}$ ions produced robust activation of the TRPV1 channel in the presence of 130 mM external Na$^+$ (*Figure 3—figure supplement 1C*), indicating that Mg$^{2+}$ can activate the TRPV1 channel directly.

Collectively, our results thus far highlight a central role of the outer pore in the allosteric control of TRPV1 channel gating. Furthermore, they suggest that the binding of DkTx, protons and external Na$^+$ to the extracellular pore of TRPV1 modulate channel function through at least partially overlapping mechanisms. We next set out to determine whether the external Na$^+$ site has an influence in the mechanism of temperature-sensitivity of the TRPV1 channel, as protons have been shown to potentiate responses of the channel to heat (*Jordt et al., 2000*), and the outer pore has been associated with temperature-sensitivity (*Myers et al., 2008*; *Grandl et al., 2010*; *Cui et al., 2012*; *Cao et al., 2014*; *Yang et al., 2014*).

## External Na$^+$ is required for activation of TRPV1 by heat

TRPV1 is widely agreed to be activated by noxious heat above 35°C in the absence of other stimuli (*Liu et al., 2003*; *Yao et al., 2010*), but whether the channel responds to lower temperatures has not been carefully examined because the open probability (P$_o$) in this temperature range is very low (*Oseguera et al., 2007*). If TRPV1 remains sensitive to changes in temperature below 35°C, it is possible that the spontaneous activation that we observed upon removing external Na$^+$ at room temperature might be caused by an alteration in the temperature-dependence for channel activation. We therefore began by investigating TRPV1 channel activity over a broad temperature range in the presence of external Na$^+$ using cells with widely varying expression levels of the channel. We measured macroscopic current-temperature (I-T) relations between ~8°C and 45°C using slow temperature ramps and observed steep temperature-dependent activation over the entire range of temperatures explored (*Figure 4A and B*). Although these I-T relations largely reflect temperature-dependent changes in P$_o$, the rate of ion conduction through open channels is also weakly temperature-dependent and thus will contribute to the slope of I-T relations. To directly evaluate temperature-dependent changes in P$_o$, we estimated the temperature-dependence of ion conduction (*Figure 4—figure supplement 1B–E*) and then calculated P$_o$-T relations from the macroscopic I-T relations (*Figure 4C* and *Figure 4—figure supplement 1F*, see Materials and methods). To qualitatively compare the slopes of P$_o$-T relations between cells and conditions, we fit a single exponential function (*Equation 1*, see Materials and methods) to the data over defined temperature ranges to obtain apparent enthalpy (ΔH$_{app}$) values (see *Figure 4—figure supplement 2C–E*). At temperatures > 35°C, we observed steep temperature-dependence to P$_o$ at positive membrane voltages (*Figure 4C and D*), which became even steeper at negative membrane voltages (*Figure 4D*, *Figure 4—figure supplement 2B and E*), consistent with previous studies (*Voets et al., 2004*; *Yao et al., 2010*). At temperatures between 8 and 25°C, a range in which the temperature-sensitivity of TRPV1 has not been previously investigated, we also observed steep temperature-dependence to P$_o$ (*Figure 4C and D*, *Figure 4—figure supplement 2C–E*). However, the entire P$_o$-T relationship could not be described by a single temperature-dependent transition due to the presence of an

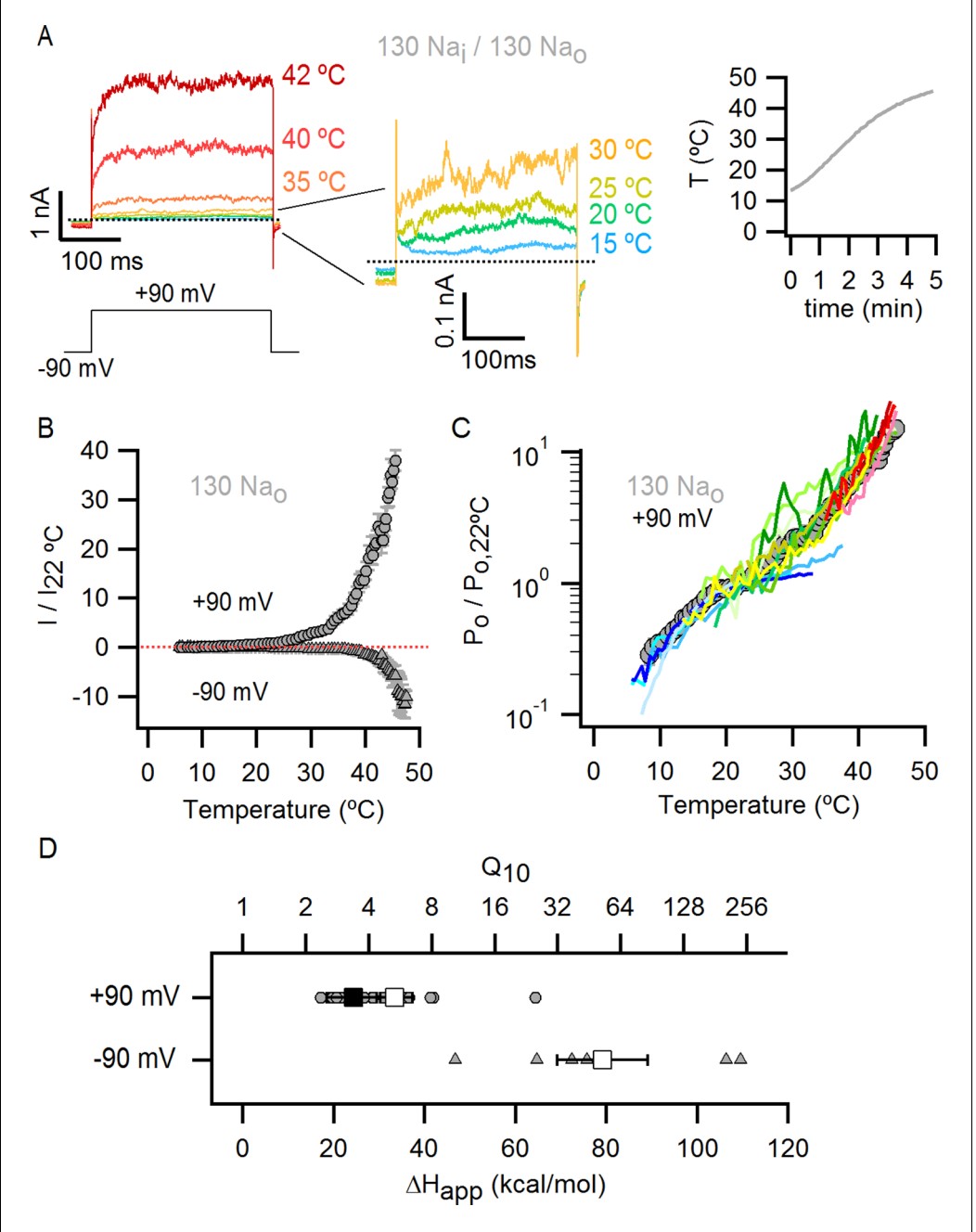

**Figure 4.** Temperature-dependent gating of TRPV1 in the presence of external $Na^+$. (**A**) Representative whole-cell current family in the presence of external $Na^+$ elicited by a train of pulses from -90 to +90 mV while increasing temperature (temperature vs time plot is shown on the right panel) using the temperature-controlled recording chamber (***Figure 4—figure supplement 1A***). The traces obtained at 15–30°C are shown at higher magnification in the middle panel. Dotted lines denote the zero-current level. (**B**) Mean current-temperature (I-T) relations obtained from experiments as in (**A**) by plotting the steady-state mean current values at -90 (triangles) and +90 mV (circles) for each voltage-pulse within a train as a function of temperature (mean ± SEM, n = 14, I-T relations for individual cells are shown in ***Figure 4—figure supplement 2A and B***). The dotted line denotes the zero-current level. (**C**) Normalized $P_o$-T relations (mean ± SEM, grey circles; individual cells are shown as colored curves, n = 14) obtained from I-T relations as in (**B**) at +90 mV as described in Methods and illustrated in ***Figure 4—figure supplement 1***. (**D**) $\Delta H_{app}$ from fits of ***Equation 1*** to $P_o$-T relations (see ***Figure 4—figure supplement 2*** for individual cells (circles) and their mean ± SEM (squares)). The mean $\Delta H_{app}$ for the fits to data with external $Na^+$ at T > 25°C ($P_o$-T relations in (**C**) colored in green, yellow, orange and red) is shown as an open square (mean ± SEM, n = 10). The mean

*Figure 4 continued on next page*

*Figure 4 continued*

$\Delta H_{app}$ from fits to data in external $Na^+$ at T < 25°C ($P_o$-T relations in (**C**) colored in blue) is shown as a closed square (mean ± SEM, n = 4). $\Delta H_{app}$ values for data at -90 mV were obtained from fits of *Equation 1* to I-T relations (***Figure 4—figure supplement 2B***), followed by subtracting the enthalpy associated with ion conduction (9 kcal/mol).

The following figure supplements are available for figure 4:

**Figure supplement 1.** Estimation of the temperature-dependence of ion conduction through an open channel for obtaining $P_o$-T relations.

**Figure supplement 2.** Individual $P_o$-T relations in the presence of external $Na^+$ measured over a wide range of temperatures uncover the presence of multiple temperature-dependent components in the gating mechanism of the TRPV1 channel.

apparent plateau near 22°C, raising the possibility that multiple temperature-dependent transitions are involved in the gating mechanism of TRPV1.

Having found that TRPV1 remains temperature-sensitive at room temperature and below in the presence of external $Na^+$, we measured macroscopic current-temperature (I-T) relations in the absence of external $Na^+$ (*Figure 5A and B*) and constructed $P_o$-T relations (*Figure 5C*). Remarkably, both I-T and $P_o$-T relations displayed greatly diminished slopes and readily detectable channel activity at the lowest experimentally accessible temperatures (*Figure 5A–C*). In addition, removal of external $Na^+$ produced marked temperature-dependent inactivation of TRPV1 channels between 25 and 38°C (*Figure 5B*, see yellow arrow), a process that typically occurs after repetitive cycles of heating and cooling in the presence of extracellular $Na^+$ (*Joseph et al., 2013*) and at temperatures above ~50°C in single heat-activation experiments (*Figure 5B*, see purple arrow). Notably, temperature-inactivated channels could not be subsequently activated by heating, but were partially responsive to saturating capsaicin (*Figure 5—figure supplement 1A*) (*Cao et al., 2014*). This facilitation of temperature-dependent inactivation in the absence of external $Na^+$ can explain why removing $Na^+$ also leads to progressive rundown of the TRPV1 channel at room temperature (*Figure 1—figure supplement 1*). We also observed that capsaicin diminishes rundown at room temperature when external $Na^+$ was removed (*Figure 5—figure supplement 1B*), consistent with a mechanistic link between temperature-dependent inactivation and channel rundown in the absence of external $Na^+$ at room temperature.

We next explored why the TRPV1 channel lacks steep temperature-dependent activation in the absence of external $Na^+$. One possibility is that the mechanism of heat activation remains intact, but that increases in $P_o$ upon heating cannot be seen because channels run down after opening during temperature ramps (see *Figure 7—figure supplement 5* and Materials and methods for an illustration using an allosteric model for channel gating). If this is the case, increasing the rate at which temperature is changed in the absence of external $Na^+$ should diminish the effects of rundown and reveal the presence of heat activation. To examine this possibility, we measured I-T relations in either the presence (*Figure 5D*, large grey circles) or absence (*Figure 5D*, large orange circles) of external $Na^+$ by rapidly jumping between solutions at different temperatures (*Figure 5—figure supplement 3*), and observed that I-T relations in the absence of external $Na^+$ obtained from temperature-ramps and from jumps agreed up to ~25°C, after which they deviated markedly (*Figure 5D*, note deviation between orange circles and small yellow circles as highlighted by the blue lines). Additionally, the rate of temperature-dependent inactivation in the absence of external $Na^+$ at T > 45°C became so fast that the speed of our temperature-jumps was not sufficient to visualize the increase in $P_o$ before significant inactivation occurred (note the last orange circle in *Figure 5D*). These results suggest that temperature-dependent activation has been greatly perturbed in the absence of external $Na^+$ at temperatures below 25°C, but that activation at higher temperatures appears largely intact (*Figure 5D*, compare orange and grey large circles). Taken together, these findings suggest that the $Na^+$ regulatory site is strongly coupled to a temperature-dependent transition that governs TRPV1 activation at low temperature, and confirm the presence of multiple temperature-dependent transitions.

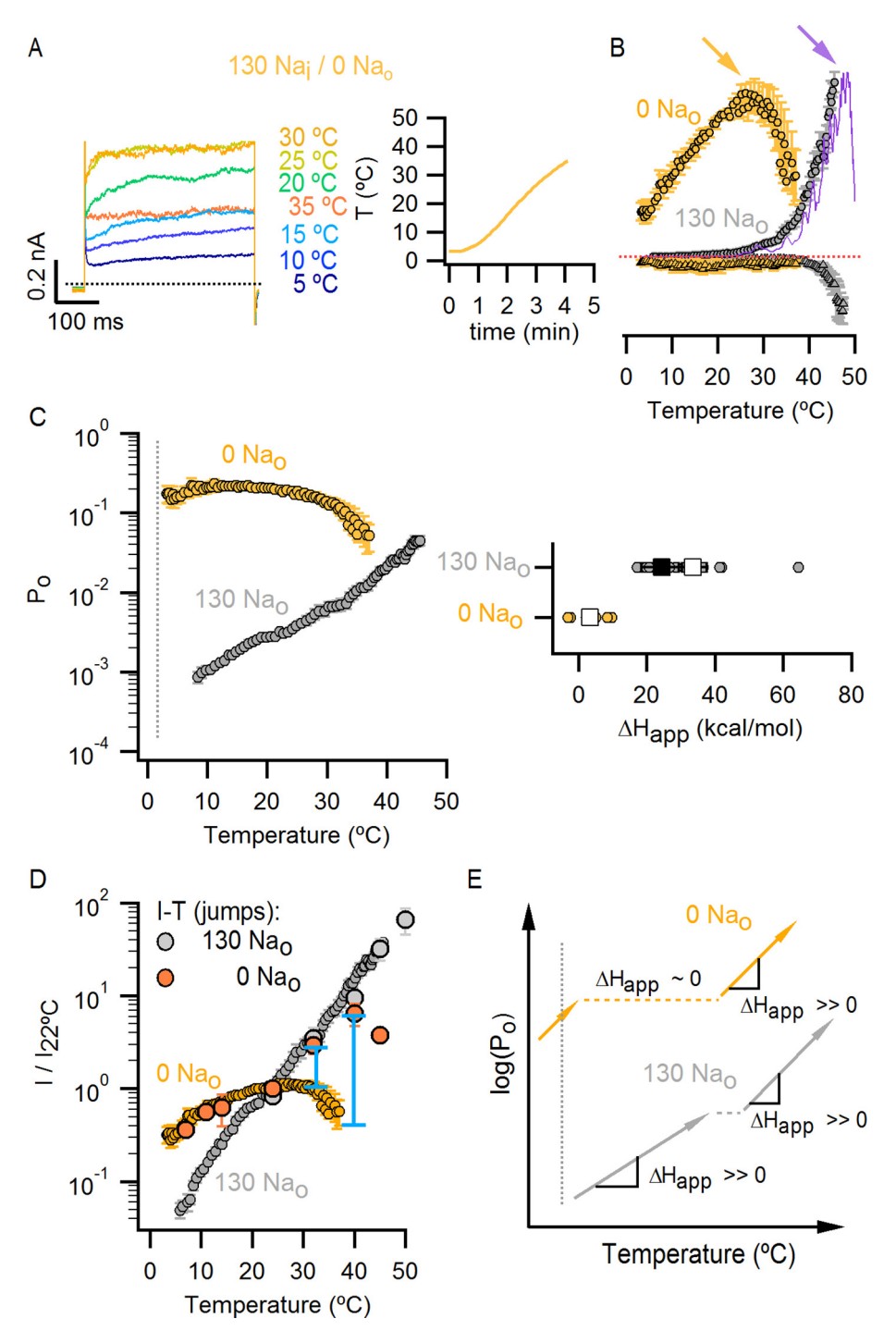

**Figure 5.** External Na$^+$ has a strong influence on temperature-dependent gating of TRPV1. (**A**) Representative whole-cell current family obtained as in *Figure 4A* in the absence of external Na$^+$. The temperature vs time plot is shown on the right panel. Dotted lines denote the zero-current level. (**B**) Mean I-T relations in the absence (obtained from data as in (**A**), mean ± SEM, n = 7) and presence (same data as in *Figure 4B*) of external Na$^+$ at +90 mV (circles) and -90 mV (triangles). The I-T relation in purple is from an experiment in the presence of 130 mM external Na$^+$ with pronounced temperature-dependent inactivation at T < 50°C. The purple and yellow arrows denote the approximate onset of inactivation for data with and without external Na$^+$, respectively. The dotted red line denotes the zero-current level. All relations are normalized to peak-current values at +90 mV. (**C**, left) Mean P$_o$-T relations (+90 mV) in the presence (grey) and absence (yellow) of external Na$^+$. P$_o$ values at room

*Figure 5 continued on next page*

*Figure 5 continued*

temperature for scaling $P_o$-T relations on an absolute $P_o$-scale were estimated from macroscopic I-V relations and noise analysis as described in Methods and *Figure 5—figure supplement 2*. The dotted vertical line delimits the lower range of experimentally accessible temperatures. $P_o$-T relations from individual cells in the absence of external $Na^+$ are shown in *Figure 6—figure supplement 1A*. (C, right) $\Delta H_{app}$ from fits of *Equation 1* (see Materials and methods) to $P_o$-T relations at +90 mV from individual cells (circles) and their mean ± SEM (squares). The mean $\Delta H_{app}$ for the fits to data with external $Na^+$ at T > 25°C is shown as an open square (mean ± SEM, n = 10). The mean $\Delta H_{app}$ from fits to data in external $Na^+$ at T < 25°C is shown as a closed square (mean ± SEM, n = 4). (D) Normalized I-T relations (+90 mV) from (B) plotted on a log-scale (small circles) with superimposed I-T relations obtained from rapid temperature-jumps from 8°C to higher temperatures (large circles, mean ± SEM, n = 3–8, see Materials and methods and *Figure 5—figure supplement 3*). The blue bars denote the increased inactivation observed in the I-T relation in the absence of external $Na^+$ obtained from slow temperature ramps relative to that obtained with rapid temperature-jumps. (E) Schematic representation of the essential features on a log-scale of $P_o$-T relations at +90 mV in the presence (grey) or absence (yellow) of external $Na^+$. The dashed lines denote plateaus in which the $P_o$ does not visibly change with temperature. Arrows denote portions of the relations in which $P_o$ steeply increases with temperature. Monoexponential fits of *Equation 1* correspond on a log-scale to straight lines with slope ~$\Delta H_{app}$.

The following figure supplements are available for figure 5:

**Figure supplement 1.** Temperature-dependent inactivation of TRPV1 channels in the absence of external $Na^+$ can be partially reversed by capsaicin.

**Figure supplement 2.** Scaling of $P_o$-T relations based on estimates of absolute $P_o$ at room temperature from macroscopic I-V relations and noise-analysis.

**Figure supplement 3.** Perfusion-mediated temperature control.

## External $Na^+$ tunes a temperature-dependent transition and stabilizes a closed state of the channel

The results obtained from measurements of $P_o$-T relations are depicted schematically in *Figure 5E*, where dotted horizontal lines represent plateau regions where steep temperature-dependence cannot be readily observed ($\Delta H_{App}$~0), and arrows represent regions where $P_o$ increases steeply with temperature ($\Delta H_{App}$>> 0). The vertical dotted grey line indicates the lowest temperature that we can achieve experimentally, with the yellow arrow crossing that temperature representing a hypothetical temperature-sensitive transition in the absence of external $Na^+$. We include this transition because the $P_o$-T relation in the absence of $Na^+$ shows some temperature-dependence to $P_o$ at T < 10°C (*Figure 5C* and *6A*). To verify the existence of this transition in the absence of external $Na^+$, we tested two manipulations that should shift the temperature range over which that transition operates. Capsazepine has been proposed to shift heat activation of TRPV1 to higher temperatures (*Matta and Ahern, 2007*), and we therefore tested whether this allosteric inhibitor might produce a rightward shift in the $P_o$-T relation, making the temperature-sensitive transition more readily detectable in the absence of external $Na^+$. Indeed, the addition of capsazepine resulted in clear temperature-dependence of $P_o$ at low temperatures in the absence of external $Na^+$ (*Figure 6A and C*). Next, we tested intermediate external $Na^+$ concentrations to see if temperature-dependent changes in $P_o$ could be detected at lower temperatures. We obtained $P_o$-T relations at 10, 30 and 65 mM external $Na^+$, and observed clear temperature-dependence at low temperatures (*Figure 6B*), with enthalpies close to those measured in the presence of 130 mM external $Na^+$ in the same temperature range (*Figure 6C*). Taken together, these findings suggest that the TRPV1 channel undergoes the same conformational transitions in response to heat regardless of whether external $Na^+$ is present, and thus that the enthalpy ($\Delta H°$) associated with $Na^+$ binding is not itself responsible for temperature-dependent gating (for a more detailed discussion see Materials and methods). Rather, it seems that external $Na^+$ modifies the midpoint (or entropy, $\triangle S$) of the temperature-sensitive transition that operates at low temperatures. Additionally, external $Na^+$ appears to stabilize a closed state of the channel independently of its effect on temperature-sensitivity, as the $P_o$ values corresponding to the

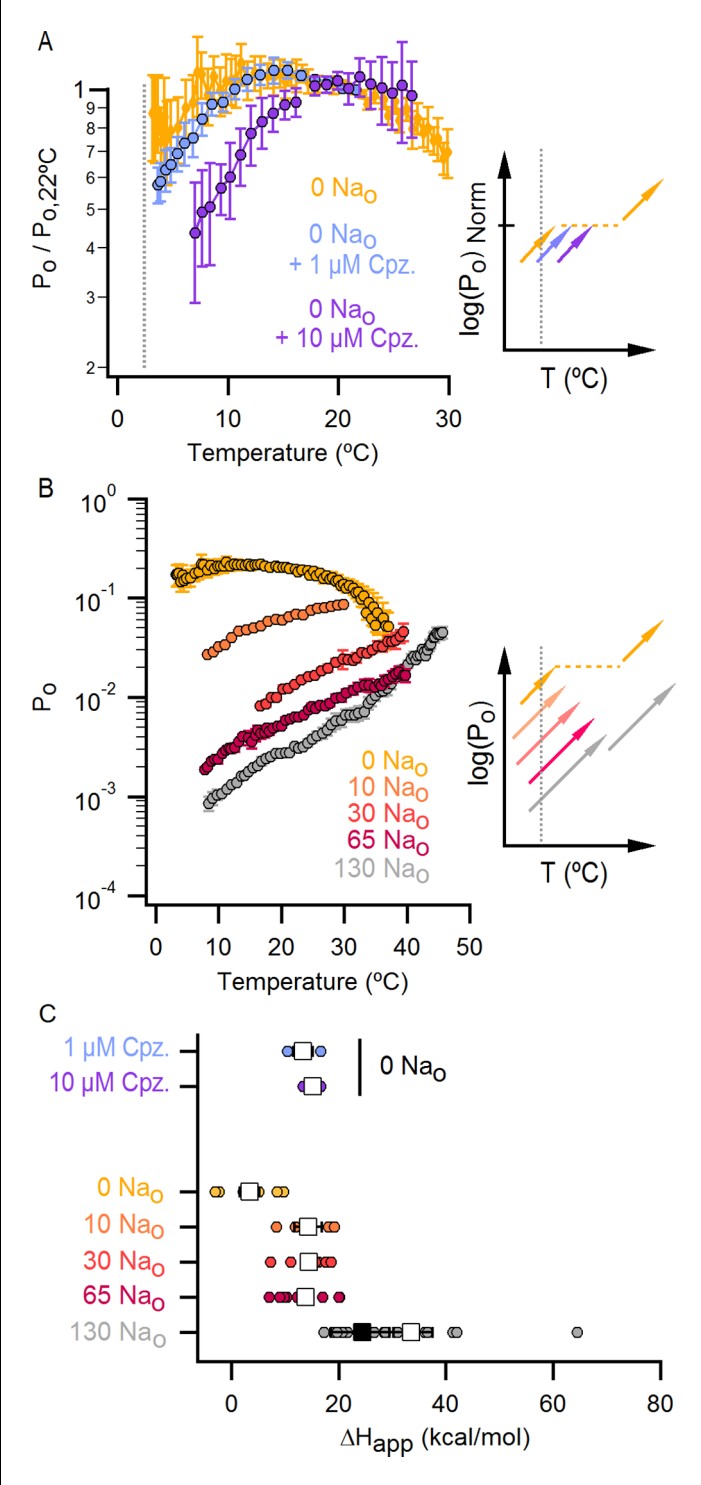

**Figure 6.** The response of the TRPV1 channel to heating is dominated by distinct conformational transitions over different temperature ranges. (**A**) Normalized $P_o$-T relations (mean ± SEM, n = 3) obtained in the presence of 0 $Na_o$ and capsazepine (Cpz). Data in 0 $Na_o$ without capsazepine are shown for comparison. The essential features of the $P_o$-T relations are schematized in the insert on the right as done in *Figure 5E*, showing that capsazepine causes a shift in the $P_o$-T relations to higher temperatures in the absence of external $Na^+$, uncovering temperature-dependent gating at low temperatures in the absence of $Na^+$. The dotted vertical line delimits the lower range of experimentally accessible temperatures. (**B**) $P_o$-T relations obtained in the presence of different concentrations of external $Na^+$ at +90 mV (mean ± SEM, n = 4–14, see *Figure 6—figure supplement 1* for

*Figure 6 continued on next page*

*Figure 6 continued*

individual cell data). $P_o$-T relations are schematized as in *Figure 5E* on the right panel insert, showing that increasing the concentration of external $Na^+$ shifts TRPV1 channel $P_o$-T relations to higher temperatures. The dotted vertical line delimits the lower range of experimentally accessible temperatures. (**C**) $\Delta H_{app}$ obtained from fits of *Equation 1* to $P_o$-T relations from individual cells (circles) and their mean ± SEM (open squares). The filled square is the mean $\Delta H_{app}$ from fits to data in 130 $Na_o$ at T < 25°C (see *Figure 4—figure supplement 2C–E*).

The following figure supplement is available for figure 6:

**Figure supplement 1.** $P_o$-T relations from individual cells obtained with different concentrations of external $Na^+$.

inferred plateau regions (denoted by dotted lines in *Figure 6B* - insert) decrease as the concentration of external $Na^+$ is increased.

## An allosteric framework for the gating of TRPV1 in response to temperature and external $Na^+$

These features of our results can be depicted in a simple allosteric model wherein two temperature-sensitive transitions are allosterically coupled to a temperature-insensitive opening transition, with $Na^+$ regulating both the midpoint ($\triangle S$) of the temperature-sensitive transition operating at low temperatures and the open/closed equilibrium (*Figure 7A*, see Materials and methods for an in-depth discussion on the choice of model and the inherent assumptions). In the presence of $Na^+$, the two temperature-sensitive transitions have similar midpoints, such that the plateau between the two is barely noticeable in the $P_o$-T relation (*Figure 6B* insert, grey arrows). Upon $Na^+$ removal, the midpoint of the first temperature-sensitive transition shifts to lower temperatures, effectively extending the middle plateau where $P_o$ is temperature-independent (*Figure 6B* insert, yellow lines and arrows) and the channel mostly occupies the open/closed equilibrium highlighted with orange squares in *Figure 7A*. Additionally, external $Na^+$ also negatively impacts the opening transition through allosteric coupling, which results in the intermediate plateau between the two temperature-dependent transitions in $P_o$-T relations having a lower $P_o$ in the presence of external $Na^+$. This simple model can qualitatively reproduce our results at intermediate $Na^+$ concentrations (*Figure 7B* and *Figure 6—figure supplement 1*). One important feature of the model is that the opening transition is relatively temperature-insensitive, and that the channel can open even when the temperature sensor is in the deactivated state (*Figure 7A,* C – O transitions with blue squares). Although we would not rule out measurable changes in $\Delta H°$ during opening, as shown by an alternate model that also satisfactorily reproduced our experimental results (*Figure 7—figure supplement 1A*), such a mechanism would require divergent temperature-sensitive transitions depending on whether external $Na^+$ is bound to the channel, which we consider to be more conceptually complex than the allosteric model proposed here (see Materials and methods for a more detailed discussion). Additionally, in the alternate model ii, external $Na^+$ influences the temperature-dependent transition that occurs at lower temperatures in the same way as in Model i (*Figure 7—figure supplement 1A*). We also cannot rule out the possibility that the temperature sensors have to activate for the channel to open (i.e., occupancy of the open states with blue squares in *Figure 7A* is negligible), as the gating of TRPV1 channels remains temperature-sensitive even at the lowest temperatures examined. However, in the absence of constraining experimental data, we have opted for the more general mechanism depicted in *Figure 7A*.

Voltage is also known to influence temperature-sensitivity of TRPV1 (*Figure 4B and D*) (*Voets et al., 2004*; *Matta and Ahern, 2007*; *Yao et al., 2010*). However, the low permeability of $NMDG^+$ precluded examination of the influence of external $Na^+$ on temperature-sensitivity of TRPV1 at negative voltages. Nonetheless, our data show that at the highly depolarized potentials where we performed our experimental studies and modeling, voltage does not have a predominant influence on temperature-sensitivity of this channel (see *Figure 7—figure supplement 3* and Materials and methods for a more detailed discussion), and therefore we did not include voltage in our modeling.

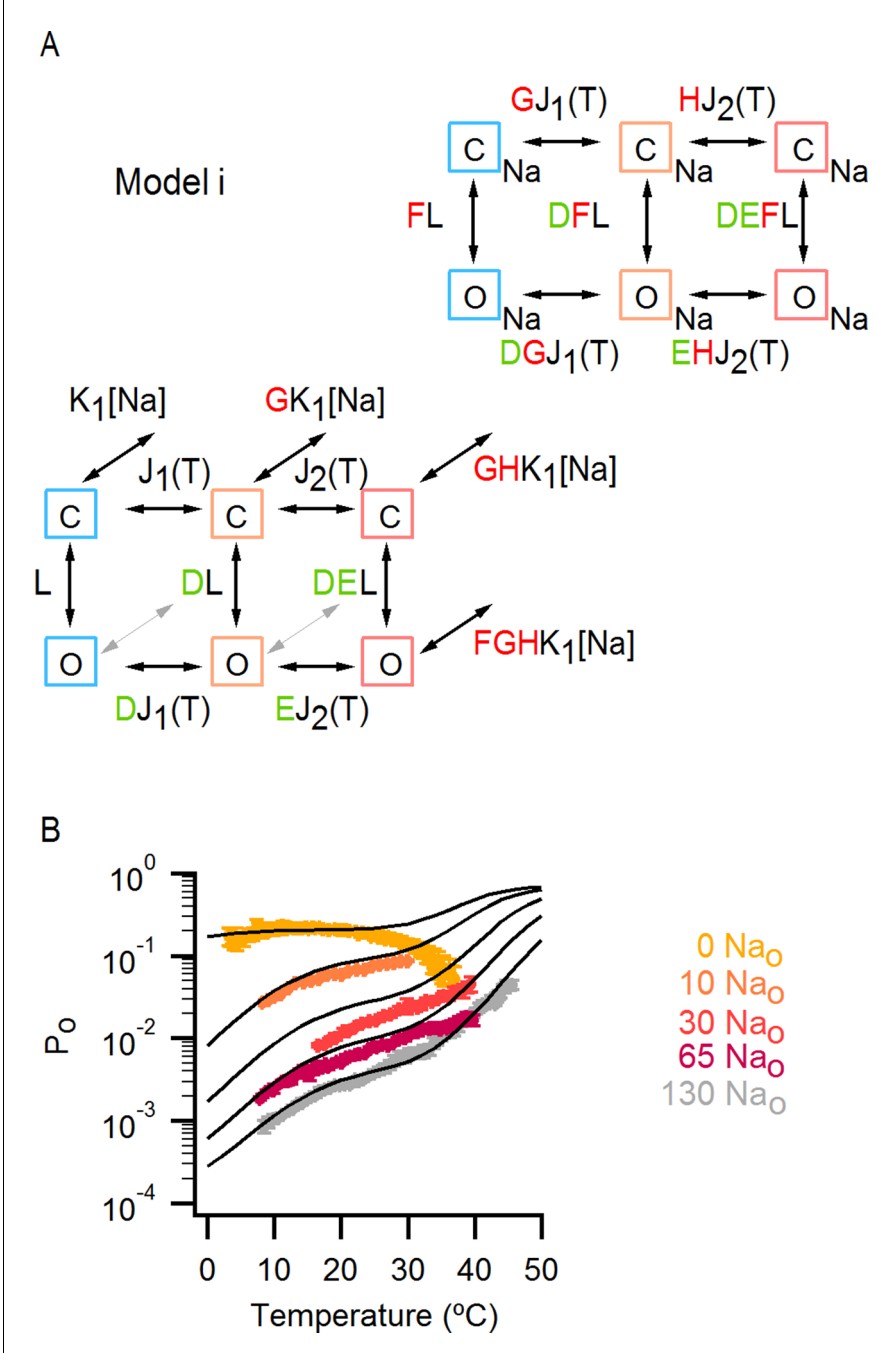

**Figure 7.** An allosteric framework for TRPV1 channel gating. (**A**) Scheme for Model i with two temperature-dependent transitions (horizontal arrows) given by equilibrium constants $J_1$ and $J_2$ of the form $J(T) = \exp(-(\Delta H° - T\Delta S°)/RT))$ and a temperature-independent opening transition (vertical arrows) with equilibrium constant L. The first and second temperature-dependent transitions promote the open state by increasing L by a factor D or E, respectively. Binding of external $Na^+$ (denoted by the subscript 'Na', diagonal arrows), which is given by equilibrium constant $K_1$, decreases $J_1$ by a factor G, $J_2$ by a factor H and L by a factor F. The color of the letters indicates a positive allosteric effect (green, allosteric factor > 1) or a negative allosteric effect (red, factor < 1) on the associated equilibrium constants. The analytical expression for $P_o$ is given in *Figure 7—source data 1A*. (**B**) Mean experimental $P_o$-T relations from data obtained in different concentrations of external $Na^+$ (colored curves, data from *Figure 6B*) with the predictions of model i superimposed (black curves). Model parameters are provided in *Figure 7—source data 2A*.

*Figure 7 continued on next page*

*Figure 7 continued*

The following source data and figure supplements are available for figure 7:

**Source data 1.** Analytical expressions for $P_o$ in different models.
**Source data 2.** Parameters for the mathematical gating models.
**Figure supplement 1.** Alternative models for describing temperature-dependent gating of TRPV1 and its modulation by external $Na^+$.
**Figure supplement 2.** High concentrations of external $Na^+$ do not fully prevent TRPV1 channel activation by positive voltages, capsaicin or heat.
**Figure supplement 3.** Assessing the influence of voltage on temperature-dependent gating of TRPV1 channels in the absence of external $Na^+$.
**Figure supplement 4.** The effect of a change in heat capacity associated with the operation of the temperature-sensor on the predictions of the allosteric gating model.
**Figure supplement 5.** Incorporating temperature-dependent inactivation into the allosteric model.

## The influence of capsaicin on temperature-dependent gating

Capsaicin is arguably the best characterized TRPV1 agonist, activating the TRPV1 channel to a maximal $P_o$ close to 1 (*Premkumar et al., 2002*; *Hui et al., 2003*), and has also been shown to influence temperature-sensitivity of this receptor (*Voets et al., 2004*; *Matta and Ahern, 2007*; *Yao et al., 2010*). We were therefore interested in determining whether it exerts a similar influence on the temperature-sensitivity of TRPV1 as external $Na^+$. To explore this possibility, we measured $P_o$-T relations using slow temperature ramps in the presence of 130 mM external $Na^+$ and different concentrations of capsaicin (*Figure 8A* and *Figure 8A—figure supplement 1A* for $P_o$-T relations from individual cells). Although we observed detectable temperature-dependent changes in $P_o$ at low temperatures for subsaturating concentrations of capsaicin, the $P_o$ values varied by more than 100-fold over the range of capsaicin concentrations, suggesting that capsaicin has a larger effect on the closed/open transition compared to the temperature-dependent transition (*Figure 8A* - insert). This contrasts with the effects of removing external $Na^+$, which produces much larger shifts in the $P_o$-T relation and smaller changes in $P_o$ (*Figure 6B*). Notably, incorporating these features in our model (*Figure 8B*) can qualitatively describe our experimental findings with capsaicin (*Figure 8A* and *Figure 8—figure supplement 1A*). The $P_o$-T relation in saturating capsaicin exhibited marginal temperature-dependent changes, consistent with a large influence of capsaicin on the open/closed equilibrium of the channel leading to full receptor activation at a saturating concentration of this agonist (*Figure 8A*).

If capsaicin has a relatively modest influence on the activation of the temperature-sensor (as compared to its larger effects on the open/closed equilibrium), then it should be possible to observe temperature-dependent changes in saturating capsaicin in a construct where capsaicin-bound channels are not fully activated, such as the turret deletion (*Figure 3E*). To test this possibility, we first determined whether TRPV1 channels lacking the pore turret are temperature-sensitive, as this is a controversial issue (*Yang et al., 2010*; *Yao et al., 2010*; *Cui et al., 2012*; *Liao et al., 2013*). Interestingly, we found that the turret deletion can still be robustly activated by heat, albeit at somewhat higher temperatures compared to the WT channel (*Figure 9A*), confirming that the turret is not the temperature sensor.

To look for temperature-dependent changes in $P_o$ for the turret-deletion construct at saturating capsaicin, we initially examined the effects of lowering temperature from 22 to 8°C on capsaicin activation in the presence of external $Na^+$ compared to maximal activation in the presence of saturating capsaicin with no external $Na^+$ (*Figure 9B*). At 8°C, the fractional activation by saturating capsaicin in the presence of external $Na^+$ was substantially lower than what we observed at 22°C, revealing that lowering temperature further decreased the efficacy of capsaicin (*Figure 9B*; 8°C colored circles; 22°C colored ramps; see black arrows). We then obtained the full $P_o$-T relationship in the presence

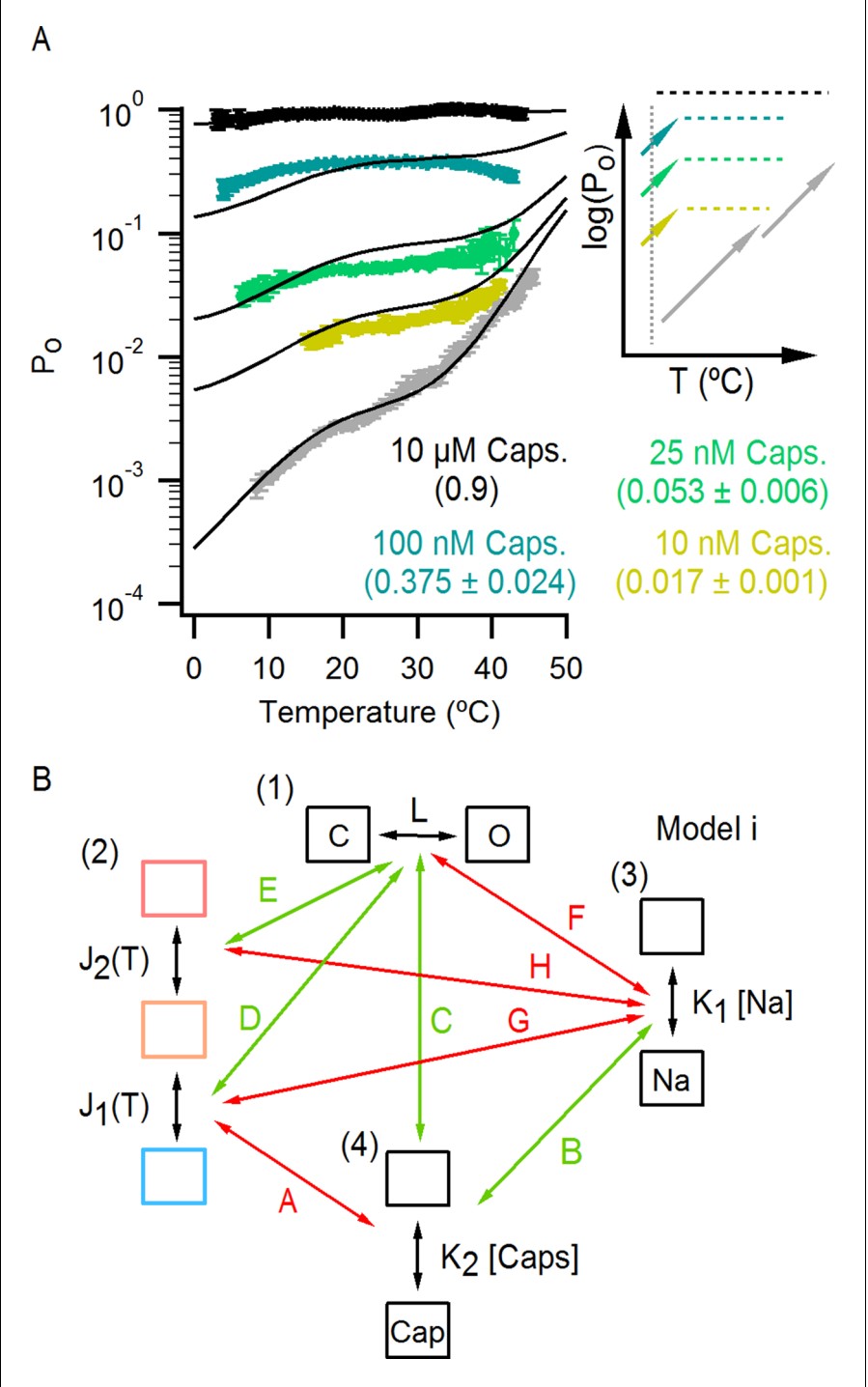

**Figure 8.** The effect of capsaicin on temperature-dependent gating of TRPV1. (**A**) $P_o$-T relations (mean ± SEM, n = 4–14) obtained in the presence of 130 mM external $Na^+$ and different concentrations of capsaicin (see *Figure 8— figure supplement 1A* for $P_o$-T relations from individual cells). The numbers in parenthesis indicate the mean $P_o$ at 22°C and +90 mV estimated from IV relations (see Materials and methods). Continuous curves are the predictions from Model i including capsaicin shown in (**B**) with $P_o$ given in *Figure 7—source data 1A* and parameters in *Figure 7—source data 2A* (see *Figure 8—figure supplement 1B* and C for theoretical $P_o$-T relations over a larger temperature range). The insert on the right shows schematized $P_o$-T relations on a log scale as in *Figure 5E*. (**B**) Schematic depiction of the allosteric Model i (*Figure 7A*) including capsaicin with four coupled equilibria denoted by the numbers in parenthesis: (1) opening and closing of the pore, determined by the temperature-independent equilibrium constant L; (2) two-step activation/deactivation of the temperature sensor(s),

*Figure 8 continued on next page*

*Figure 8 continued*

given by temperature-dependent equilibrium constants $J_1$ and $J_2$ of the form $J(T) = e^{-\frac{\Delta H^o - T\Delta S^o}{RT}}$; binding/unbinding of sodium (3) or capsaicin (4), determined by temperature-independent equilibrium constants $K_1$ and $K_2$, respectively. Arrows indicate allosteric coupling between equilibria, with red denoting allosteric inhibition (allosteric factor < 1) and green stimulation (allosteric factor > 1). The magnitude of the coupling is determined by the multiplicative temperature-independent allosteric factors next to the arrows.

The following figure supplement is available for figure 8:

**Figure supplement 1.** $P_o$-T relations from individual cells for different concentrations of capsaicin and predictions of the allosteric model over an extended range of temperatures.

of external $Na^+$ and capsaicin, as before for the WT channel, but for the turret deletion we observed large temperature-dependent changes in $P_o$, with two components in the relation separated by a clear plateau at intermediate temperatures (*Figure 9C* and *Figure 9—figure supplement 1B* for $P_o$-T relations from individual cells). These findings with the turret deletion further support the allosteric framework for conceptualizing the mechanism by which $Na^+$ and capsaicin influence temperature-dependent gating. Notably, we could accurately describe the $P_o$-T relations in saturating capsaicin and the dose-response for external $Na^+$ for the turret deletion using model i (*Figure 9—figure supplement 1A and B*), simply by assuming that the deletion of the pore turret increased the affinity of the channel for external $Na^+$ and reduced the strength of the coupling between the binding of capsaicin and the opening of the pore (*Figure 7—source data 2E*). Together, these results qualitatively show that capsaicin influences both the activity of the temperature-sensor and the open/closed equilibrium, but appears to have a larger influence on the latter. Even though the values of the coupling parameters in our modeling are not well-determined, we believe that the larger influence of capsaicin on the open/closed equilibrium is robust enough to support this conclusion.

## Influence of DkTx on temperature-dependent gating

The results thus far demonstrate that the external pore of TRPV1 contains $Na^+$ ion binding sites that exert strong control over temperature-dependent activation and inactivation, as well as the closed/open equilibrium. Our previous results with DkTx suggest external $Na^+$ and DkTx modulate the TRPV1 through a common mechanism (*Figure 3B–D*). In addition, our recent analysis of the interaction between DkTx and TRPV1 suggests that the toxin activates the channel by inducing a displacement of the S5-S6 segments relative to the S1-S4 domain (*Figure 10—figure supplement 1A*), which results in the disruption of a cluster of buried hydrophobic residues at the extracellular ends of the S5 and S6 segments (*Bae et al., 2016*). This is interesting in light of a recent demonstration in the Shaker Kv channel that movement of hydrophobic residues between buried and solvent-exposed environments imparts temperature-dependence (*Chowdhury et al., 2014*). We therefore measured $P_o$-T relations in the presence of external $Na^+$ and DkTx to test whether the binding of the toxin has an effect on temperature-sensitivity, and observed that activation of the channel by heat was effectively ablated between 5 and 35°C (*Figure 10,* and *Figure 10—figure supplement 1B* for $P_o$-T relations from individual cells). This finding is particularly striking when considering that the $P_o$ is situated well below the theoretical upper limit of 1.0. These results demonstrate that DkTx also exerts strong control over temperature-sensor activation, which in our model can be achieved simply by trapping the channel in the intermediate plateau and preventing temperature-sensor deactivation or full activation (*Figure 10,* insert).

## Discussion

The present results examining the regulation of temperature-dependent gating by external $Na^+$ and capsaicin have four important implications for understanding the gating mechanisms of the TRPV1 channel. First, our results demonstrate that the TRPV1 channel has evolved external $Na^+$ binding sites that must be occupied for the channel to remain closed at physiological body temperatures in mammals. The steeply temperature-sensitive transition that normally opens TRPV1 in response to noxious heat still operates in the absence of $Na^+$, yet without external $Na^+$ the channel would be

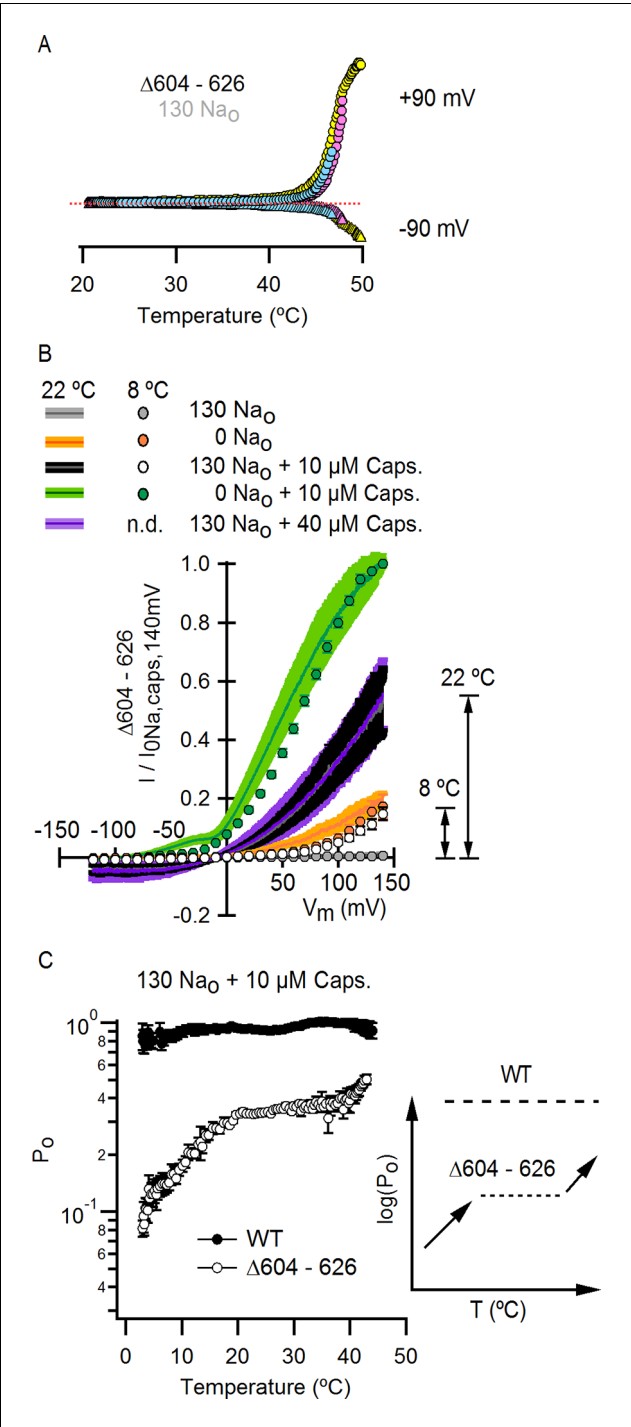

**Figure 9.** Deletion of the outer pore turret uncovers temperature-sensitive gating of TRPV1 in saturating capsaicin. (A) Three representative TRPV1 Δ604–626 whole-cell I-T relations measured in 130 mM external Na$^+$ in response to pulses from -90 to +90 mV as in *Figure 4A* and B. The dotted line is the zero-current level. (B) Normalized I-V relations for TRPV1 Δ604–626 obtained in the whole-cell configuration at 22°C (continuous curves are the mean, lighter envelopes the SEM, n = 6, same data as *Figure 3E*) or 8°C (circles, mean ± SEM, n = 9) in response to voltage ramps or families of voltage pulses, respectively. The arrows indicate the fractional activation by capsaicin in the presence of external Na$^+$ at both 8 and 22°C. (C) P$_o$-T relation (mean ± SEM, n = 12) for TRPV1 Δ604–626 (open circles) at +90 mV scaled based on the I-V relations in (B) at 22°C (assuming maximal P$_o$ of 0.9 at room temperature in the presence of 10 μM capsaicin without external Na$^+$). The mean P$_o$-T relation for the WT channel in the presence of 130 mM external Na$^+$ and 10 μM capsaicin is shown as closed black circles (data from *Figure 9 continued on next page*

*Figure 9 continued*

*Figure 4—figure supplement 1B*, mean ± SEM, n = 11). The insert on the right schematizes as in *Figure 5E* the $P_o$-T relations for both WT (long-dash line) and Δ604–626 (short-dash line and arrows) channels in the presence of saturating capsaicin and external $Na^+$. See *Figure 9—figure supplement 1B* for $P_o$-T relations from individual cells.

The following figure supplement is available for figure 9:

**Figure supplement 1.** Influence of the pore turret of TRPV1 on temperature-dependent gating and modulation by sodium.

spontaneously open at physiological temperatures, many channels would be inactivated, and responses to noxious heat would therefore be diminished. Our results also show that the $Na^+$ sites will play physiologically important roles in activation of the channel by acid, as protonation of acidic residues involved in binding $Na^+$ leads to diminished $Na^+$ occupancy and activation of the channel. This connection between $Na^+$ and pH regulation provides a mechanistic explanation for previous observations where mildly acidic solutions potentiate activation of TRPV1 by heat (*Jordt et al., 2000*). Thus, this $Na^+$ regulatory mechanism plays a fundamental role in tuning the properties of TRPV1 to serve as a detector of noxious heat and acid, and possibly other stimuli that have yet to be examined.

Second, our results reveal the existence of multiple temperature-sensitive transitions that are coupled to the opening transition (see model in *Figure 7A* ). Although we see only hints of the presence of distinct temperature-dependent transitions in the presence of external $Na^+$ (*Figure 4C*), they become unambiguous when $Na^+$ is removed (*Figure 5D*)and when studied in the presence of

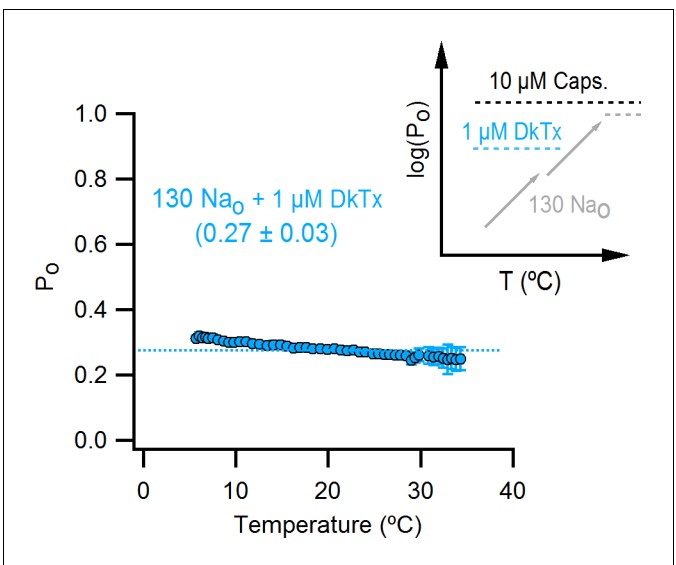

**Figure 10.** The binding of DkTx to the outer pore of TRPV1 effectively ablates temperature-dependent gating over a wide range of temperatures. $P_o$-T relation (mean ± SEM, n = 9) for WT TRPV1 obtained in the presence of external $Na^+$ and DkTx. The dotted line denotes the mean $P_o$ at 22°C and +90 mV as estimated from the I-V relations in DkTx relative to saturating capsaicin shown in *Figure 3C*. The insert to the upper right represents experimental $P_o$-T relations on a log scale measured in the presence of external $Na^+$ and saturating capsaicin (dashed black line denoting the absence of temperature-dependent gating), external $Na^+$ and DkTx (dashed blue line, no temperature-dependent gating) and external $Na^+$ alone (two temperature-dependent transitions). See *Figure 10—figure supplement 1B* for $P_o$-T relations from individual cells.

The following figure supplement is available for figure 10:

**Figure supplement 1.** DkTx binding to TRPV1.

capsaicin with the turret deletion construct (*Figure 9C*). Our results suggest that the role of the first temperature-sensitive transition is to maintain the channel in a closed state at physiological temperatures, whereas the second transition functions to activate the channel in response to noxious heat. It is interesting that both $Na^+$ and capsaicin do not seem to have a substantial impact on the second transition, at least to the extent that we could study this component in the presence of inactivation. In contrast, external $Na^+$ exerts strong control over the first transition. Capsaicin also modifies the shape of macroscopic $P_o$-T relations (*Figure 8A*), but we were able to detect temperature-dependent transitions at low temperatures even in the presence of near-saturating capsaicin concentrations (e.g. 100 nM) that increase the $P_o$ at room temperature to values higher than those measured in the absence of external $Na^+$ or in the presence of DkTx (*Figure 3C*). In contrast, we observed marginal to no temperature-dependence in the $P_o$-T relations at low temperatures in the absence of external $Na^+$ or in the presence of DkTx (*Figure 5D* and *Figure 10*). These observations suggest that the vanilloid has a larger influence on the open/closed equilibrium than on the temperature-sensing machinery. The existence of multiple temperature-sensitive transitions also provides a possible explanation for why $P_o$-T relations at negative voltages are at least two-fold steeper than at positive voltages (*Figure 4B*) (*Voets et al., 2004*; *Yao et al., 2010*). If hyperpolarization shifts the temperature-sensitive transition operating at lower temperatures so as to coincide with the second temperature-sensitive transition, the $\triangle H^o$ of the two transitions would add to make the $P_o$-T steeper. Overall, the present results on the influence of $Na^+$ and capsaicin on temperature-dependent gating in TRPV1 provide a set of essential constraints for understanding the gating mechanism of the channel using an allosteric framework.

Third, our results provide clues about where these important $Na^+$ binding sites are located. Because $Na^+$ inhibition has only very weak voltage-dependence, these $Na^+$ binding sites are likely to be located outside the ion permeation pathway within each subunit. Indeed, the large outward $Na^+$ currents we observe on removing external $Na^+$ at positive voltages indicate that the permeation pathway must have a high $Na^+$ occupancy even in the absence of external $Na^+$. Mutating acidic residues forming the external $Na^+$ site would weaken $Na^+$ binding affinity, possibly leading to lower $Na^+$ occupancy in the presence of 130 mM external $Na^+$. However, such an effect would not necessarily cause spontaneous TRPV1 activity because the lower $Na^+$ occupancy would be expected to promote the form of inactivation that we observed on removing external $Na^+$. Additionally, removing external $Na^+$ would be expected to cause less activation of the channel compared to the WT protein because ion occupancy would be diminished by the mutant. E600 in the extracellular loop just after the S5 helix (*Figure 3—figure supplement 1A*) has been proposed to be responsible for proton-mediated potentiation of heat- and capsaicin-dependent activation of the channel (*Jordt et al., 2000*), making it an excellent candidate for forming the $Na^+$ binding site. The E600Q mutant displayed modest spontaneous activity when recording at room temperature (*Ohta et al., 2008*), and only weak activation when removing external $Na^+$ or applying DkTx when compared with the WT channel, consistent with a role in forming the $Na^+$ binding site. Interestingly, E600 is located near the pore turret (*Liao et al., 2013*) (*Figure 3—figure supplement 1A*), which when deleted from TRPV1 causes a dramatic increase in the apparent affinity of $Na^+$ (*Figure 3F*), and this residue is also located very close to where DkTx binds (*Cao et al., 2013*) (*Figure 3—figure supplement 1A* and *Figure 10—figure supplement 1A*).

Finally, our results have important implications for interpreting the recent cryo-EM structures of TRPV1 and in localizing the region of the protein involved in temperature sensing. In the cryo-EM studies on the turret deletion construct of TRPV1, capsaicin appears to produce a partial opening of the internal pore, and the combined influence of RTx and DkTx to produce further expansion of the internal pore together with a conformational change within the selectivity filter (*Cao et al., 2013*) (*Figure 10—figure supplement 1A*). These differences between the two structures might indicate that there are two gates that can be differentially regulated by activating stimuli (*Cao et al., 2013*) (*Figure 1A*). However, we discovered that capsaicin is a partial agonist of the turret deletion construct, suggesting that the partially open features of the capsaicin-bound channel may not represent a unique conformation of the protein, but an average of closed and open states. The cryo-EM structure of the apo state of TRPV1 (*Liao et al., 2013*), however, reveals that the internal pore must expand for the pore to support ion permeation, consistent with accessibility studies (*Salazar et al., 2009*). The observation of temperature-independent plateaus in $P_o$-T relations under diverse conditions that produce submaximal open probabilities suggests that the opening of the pore does not

require the two temperature-dependent transitions to occur, and that the open/closed equilibrium itself is not the source of the temperature-sensitivity of TRPV1. If opening of the internal pore is indeed temperature-insensitive, what is the significance of the structural changes in the external pore observed in the structure of the RTx/DkTx-bound channel? The present findings with external $Na^+$ and DkTx suggest that these structural changes in the external pore are tightly associated with the temperature-sensing machinery, either by participating in allosteric coupling or in the actual mechanism of temperature-sensing.

## Materials and methods

### Channel constructs, cell culture and transfection

The WT rat TRPV1 channel in pcDNA3.1 vector was kindly provided by Dr. David Julius (UCSF), and subcloned into the low-expression pcDNA1 vector. The E600Q point mutation and Δ604–626 deletion were introduced into rTRPV1 using a two-step PCR mutagenesis technique and the resulting constructs were verified by sequencing.

HEK293 cells were cultured following standard protocols (*Li et al., 2015*) and transiently transfected with plasmids for the expression of TRPV1 and GFP (pGreen-Lantern, Invitrogen, Carlsbad, CA) using FuGENE6 (Promega, Madison, WI).

### Electrophysiology

Standard whole-cell patch clamp recordings from transiently transfected HEK293 cells at room temperature (22–24°C) were performed unless stated otherwise. The data were acquired with an Axopatch 200B amplifier (Molecular Devices, Sunnyvale, CA), filtered with an 8-pole low-pass Bessel filter (model 900, Frequency Devices, Ottawa, IL) and digitized with a Digidata1322A interface and pClamp10 software (Molecular Devices). All data were analyzed using Igor Pro 6.34A (Wavemetrics, Portland, OR). Pipettes were pulled from borosilicate glass and heat-polished to final resistances between 1 and 5 MΩ. 80–95% series resistance ($R_s$) compensation was used in all whole-cell recordings except those involving changes in temperature.

The intracellular recording solution consisted of (in mM): 130 NaCl, 10 HEPES, 10 EGTA, pH 7.4 (NaOH/HCl). For experiments involving changes in temperature, 10 mM $MgCl_2$ was added to the intracellular solution to block endogenous inward-rectifying and low-threshold temperature-sensitive currents that were observed in some recordings. The addition of the divalent did not cause any appreciable effect on TRPV1 channel currents, consistent with a previous report (*Yang et al., 2014*). Moreover, we did not observe significant contaminating currents at room temperature when magnesium was omitted from the pipette solution, except in a small subset of cells that exhibited noticeable inward currents that were easily recognizable due to the outward-rectifying character of TRPV1, and thereby could be readily discarded. The control extracellular solution (130 $Na_o$) consisted of (mM): 130 NaCl, 10 HEPES, 10 EGTA, pH 7.4 (NaOH/HCl). The zero-sodium extracellular solution (0 $Na_o$) consisted of (mM): 130 NMDGCl, 10 HEPES, 10 EGTA, pH 7.4 (NMDG/HCl). For solutions with different extracellular $Na^+$/$NMDG^+$ concentrations, NaCl was replaced with NMDGCl. Capsaicin and capsazepine (100 mM) stock solutions were prepared in ethanol or 1:1 ethanol/DMSO, respectively. All chemicals and reagents were from Sigma-Aldrich (San Luis, MO). DkTx was produced recombinantly, folded and purified as described previously (*Bae et al., 2012*).

A holding potential of -90 mV was used in all experiments. The data were acquired at 5–10 kHz and low-pass filtered at 2 kHz. For voltage-ramps, voltage was initially stepped down to -120 mV for 50 ms, then ramped up to +140 mV over 1 s before returning to -90 mV. Ramps were applied every 5 s. For I-V relations obtained with steps, voltage was stepped from -90 mV to the test potential for 100 (room temperature recordings) or 400 ms (low temperature recordings, due to slow kinetics), and returned to -90 mV after each pulse. Test pulses went from -120 to +140 mV in 10-mV increments. Current time courses at a fixed voltage were obtained by applying trains of pulses from -90 to +90 mV of either 100 (experiments at room temperature) or 300 ms duration (experiments starting at lower temperatures). The pulse interval was varied, using 3–5 s for I-T relations and 100 ms for time courses at room temperature.

In all experiments at room temperature, a gravity-fed rapid solution exchange system (RSC-200, BioLogic, France) was used. The cells were lifted from the coverslip and placed in front of glass

capillaries perfused with different solutions. For I-V relations at a low temperature and fast temperature-jumps, a custom-made temperature-controlled perfusion system was used, as described below and in *Figure 5—figure supplement 3A*. I-T relations in response to slow temperature ramps were obtained in a temperature-controlled microincubator stage (see below and *Figure 4—figure supplement 1A*).

$P_o$-T relations were constructed by factoring out the temperature-dependence of conduction through an open channel from I-T relations (see below), and then scaled based on the relative current magnitudes at each condition measured at +90 mV and at room temperature from voltage-ramps (see below for details). The scaling of $P_o$-T relations was also verified using noise analysis (see below).

To obtain a qualitative assessment of the temperature-dependence of channel function, the steepest portions of I-T relations were fit to *Equation 1*:

$$I(T) = \exp\left(-\frac{\Delta H_{app} - T\Delta S_{app}}{RT}\right) \tag{1}$$

where I(T) is the current as a function of temperature, $\Delta H_{app}$ and $\Delta S_{app}$ are the apparent enthalpy and entropy, respectively, T is the temperature (in Kelvins) and R is the gas constant. Approximate $Q_{10}$ values were calculated from $Q_{10} \approx e^{\frac{\Delta H_{app}}{20}}$.

All dose-response curves were obtained with voltage-ramps and fit to the Hill equation: $\frac{I}{I_{max}} = I_{min} + \frac{I_{max} - I_{min}}{1 + \left(\frac{K_{1/2}}{[X]}\right)^s}$, where $I_{min}$ and $I_{max}$ are the currents at saturating concentrations of $Na^+$ and 0 $Na^+$, respectively, $K_{1/2}$ is the concentration of half-maximal inhibition, s is the Hill coefficient and [X] is the molar concentration of $Na^+$. Dose-response relations for extracellular $Na^+$ at pH 7.4 and 5.5 were obtained using 130 mM internal $Na^+$ and, for all test solutions, a total concentration of external monovalent cations ($Na^+$ + $NMDG^+$) of 130 mM. For the dose-response relation at pH 6.0, 300 mM intracellular $Na^+$ was used with a total concentration of external cations of 300 mM. For experiments at pH 6 and 5.5, methanethiosulfonic (MTS) acid was used as a counterion instead of chloride to ablate endogenous proton-activated chloride currents (*Lambert and Oberwinkler, 2005*). Solutions at pH 6.0 or 5.5 were buffered with 10 mM Bis/Tris or 10 mM MES, respectively.

## Obtaining and analyzing macroscopic I-T relations using slow temperature ramps

For obtaining I-T relations a custom-modified temperature-controlled microincubator stage was used (*Figure 4—figure supplement 1A*). Both T and I were simultaneously recorded and the bath was kept static during the course of the recordings. Only a single experimental solution was tested per cell. Currents were elicited while increasing or decreasing temperature by applying trains of voltage pulses from -90 to +90 mV (from a holding of -90 mV), with pulse durations that ranged from 100 to 400 ms. Longer pulses were used for I-T relations at the lowest temperatures, since activation kinetics were much slower. Pulse length did not have an appreciable effect on I-T relations when evaluated at the same time-points for experiments with different pulse durations (data not shown). For obtaining I-T relations extending to cold temperatures, experiments were started at room temperature, and the temperature was decreased to the lowest value in which TRPV1 currents could still be reliably measured, as indicated by time-dependent relaxations in response to the depolarizing pulses that were at least twice the magnitude of the currents at -90 mV. Immediately after reaching the minimal temperature, experiments were initiated by increasing the temperature. For I-T relations starting at room temperature, the experiments were started without an initial cooling phase.

To generate I-T relations, steady-state currents at +90 mV for each pulse within a train were quantified and plotted as a function of temperature. Since not all individual I-T relations obtained for a given experimental solution contained data at exactly the same temperatures, currents at similar temperatures from different cells were manually matched before averaging both the currents and the temperatures. Individual I-T relations were normalized to their current value at room temperature, and then averaged to obtain mean I-T curves. Standard errors for the temperature are also included in all I-T and $P_o$-T relations shown, but are too small to be seen. No series resistance ($R_s$) compensation was used in the I-T experiments, since $R_s$ changes with temperature, leading to

oscillations and patch rupture during the course of the experiments. To minimize voltage errors, only currents less than 2.5 nA were used for analysis.

## I-T relations obtained from fast perfusion-mediated temperature changes

Solutions were passed through glass capillary spirals immersed in water baths at different temperatures, and recordings were performed in a small-volume (200–500 µL) chamber during constant perfusion (*Figure 5—figure supplement 3A*). Temperature was measured with a thermistor located very close to the pipette tip. One perfusion line was used for a given test solution and temperature. An intracellular solution containing 130 mM $Na^+$ + 10 mM $Mg^{2+}$ was used. All experiments started in 130 $Na_o$ at 8°C (*Figure 5—figure supplement 3B*), followed by 8°C in the absence of external $Na^+$ (i.e. 130 $NMDG_o$) to provide a reference point for normalization before changing to a test solution with or without external $Na^+$ and at T $\geq$ 11°C. A single test solution was evaluated per cell, and test solutions with and without external $Na^+$ were analyzed separately to generate the I-T relations shown in *Figure 5D* (grey and orange large circles). In the representative experiments shown in *Figure 5—figure supplement 3B*, the test solutions were 0 $Na_o$ at 32°C (left panel) and at 22°C (right panel). In some cases, saturating capsaicin at room temperature was applied at the end of the experiment. For each cell, the current measured in the test solution was normalized to the current in control solution (i.e. 0 $Na_o$, 8°C), and the resulting ratio averaged among experiments obtained using the same test condition (both temperature and presence or absence of external $Na^+$). An I-T relation (either with or without external $Na^+$) was finally constructed by plotting together the mean current ratios as a function of the temperature of the test solution. The final I-T relations in *Figure 5D* were scaled so that the current ratio at room temperature (22°C) was equal to 1.

## Estimation of the temperature-dependence for ion-conduction through an open channel

Temperature-dependent changes in macroscopic currents have two predominant sources: the temperature-dependence of channel gating ($P_o$(T), opening and closing of the pore) and the rate of ion conduction through an open channel i(T). The contribution of the leak currents was considered negligible since the magnitude of the measured currents was always considerably larger than the magnitude of the leak, and accounting for the leak did not significantly change the shape of the I-T or $P_o$-T relations under any condition (data not shown). To estimate the temperature-dependence of ion conduction through an open channel we measured I-T relations in the presence of a saturating concentration of capsaicin, a condition in which all channels remain open over the experimental temperature range. I-T relations in saturating capsaicin (*Figure 4—figure supplement 1B*, closed circles) could be satisfactorily fit by a single exponential function of temperature ($eq. 1 - i(T) = i_0 \times e^{\frac{\Delta H^{\neq}}{RT}}$) over the studied temperature range, indicating that $P_o$ does not change appreciably from 7–40°C. This is also consistent with G-V relations obtained from families of voltage pulses at room temperature (*Figure 1—figure supplement 3A*) and at 8°C (*Figure 7—figure supplement 3A*) in saturating capsaicin with 130 mM extracellular $Na^+$, which superimpose very well and show clear saturation at positive potentials (*Figure 7—figure supplement 3B*). Individual I-T experiments were normalized by their own fits to *Equation 1*, and the resulting data were normalized to their respective values at 24°C before averaging, to finally generate a curve that only reflects changes in $P_o$ as a function of temperature (*Figure 4—figure supplement 1B*, open circles). As expected, the resulting normalized $P_o$-T relation for saturating capsaicin does not exhibit large temperature-dependent changes. The mean enthalpy for ion conduction estimated this way (9.0 ± 0.5 kcal/mol, *Figure 4—figure supplement 1B*, insert) was used to factor i(T) out of all other macroscopic I-T relations (see *Figure 4—figure supplement 1F* for an example of an I-T to $P_o$-T transformation for data obtained with 130 mM external $Na^+$). Single-channel current amplitude measurements at +90 mV with and without external $Na^+$ measured at several temperatures (*Figure 4—figure supplement 1C and D*) could be satisfactorily fit by *Equation 1* with an enthalpy of 9 kcal/mol (*Figure 4—figure supplement 1E*, see below for experimental details) and our estimate of 9 kcal/mol for the temperature-dependence of i(T) is close to previous estimates for TRPV1 (*Liu et al., 2003*; *Grandl et al., 2010*) and other ion channels (*Hille, 2001*).

## Single-channel current amplitude measurements at several temperatures

Data for single-channel recordings were acquired at 10 kHz and filtered at 2 kHz using pipettes with resistances of 5–14 MΩ. Pipettes were covered in Q-dope or dental wax. Currents were elicited by 500-ms pulses from -90 to +90 mV every 100 ms during heating-temperature ramps in outside-out patches containing a few channels (*Figure 4—figure supplement 1C and D*). Temperature was controlled with the micro-incubator stage (*Figure 4—figure supplement 1A*). Traces without channel openings were used for leak subtraction. All-points histograms for data obtained within a 2°C interval were assembled from segments of recordings where single-channel openings were clearly discernible, and data were fitted to a sum of Gaussian functions to obtain the mean current value for a single open channel (*Figure 4—figure supplement 1E*).

## Scaling of $P_o$-T relations based on macroscopic current recordings

Once the temperature-dependence of ion conduction for an open channel has been factored out of I-T relations, the resulting data reflect only the changes in $P_o$ as a function of temperature. To graph the resulting relations on an absolute $P_o$ scale, we used the relative magnitudes of macroscopic currents at room temperature measured with the different solutions tested in this study. I-V relations in saturating capsaicin with and without external $Na^+$ superimpose at positive potentials (*Figure 1C*), we did not observe large changes in single channel conductance at +90 mV and at room temperature between patches with or without external $Na^+$ (*Figure 2A*), and capsaicin does not affect the single-channel conductance of TRPV1 (*Premkumar et al., 2002*; *Hui et al., 2003*), suggesting that the single channel current at positive voltages is not substantially altered under our experimental conditions. Based on measurements provided in the literature using single-channel recordings (*Premkumar et al., 2002*; *Hui et al., 2003*) or noise-analysis of macroscopic currents (*Oseguera et al., 2007*), we assumed that the $P_o$ in saturating capsaicin (10 μM) + 130 mM external $Na^+$ at +90 mV and at room temperature is ~0.9. The $P_o$ at room temperature and +90 mV for 0 $Na_o$ (i.e., 130 mM extracellular NMDG) was then estimated from the macroscopic I-V relations (*Figure 5—figure supplement 2A and C*) to be ~0.2. For other experimental conditions where $P_o$ is similar to that measured in 0 $Na_o$, we estimated the $P_o$-values at room temperature and +90 mV simply by recording I-V relations at the test conditions and 0 $Na_o$ and scaling based on 0 $Na_o$. The following conditions were scaled in this way (*Figure 5—figure supplement 2A and C*): 0 $Na_o$ + 1 μM Cpz., 0 $Na_o$ + 10 μM Cpz., 10 $Na_o$, 30 $Na_o$, 130 $Na_o$ + 100 nM caps., 130 $Na_o$ + 25 nM caps., 130 $Na_o$ + 1 μM DkTx. For experimental conditions where $P_o$ was much lower than for 0 $Na_o$, we used 30 $Na_o$ as a reference in cells expressing TRPV1 to higher levels (*Figure 5—figure supplement 2B and C*). The following conditions were scaled in this way: 130 $Na_o$, 65 $Na_o$, 130 $Na_o$ + 10 nM caps. Notably, the $P_o$ value estimated here for 130 $Na_o$ is consistent with measurements of unliganded $P_o$ for TRPV1 at depolarized potentials (*Oseguera et al., 2007*). We also validated our scaling procedure using noise analysis (see below for methodological description). Using this approach, we obtained a mean $P_o$ at ~24°C of 0.91 ± 0.02 (mean ± SEM, n = 4) for data in 130 $Na_o$ + 10 μM caps. and a $P_o$ of 0.21 ± 0.04 in the absence of external $Na^+$ and capsaicin (see *Figure 5—figure supplement 2G* for $P_o$ values from individual cells at different temperatures), which is very close to our initial estimated values.

It is important to note that our estimated absolute $P_o$ values from macroscopic recordings in HEK293 cells in the presence of external $Na^+$ at elevated temperatures are almost an order of magnitude lower than those measured previously using single-channel recordings from oocytes (*Liu et al., 2003*), which showed that the $P_o$ at T > 50°C is close to 1. Macroscopic recordings from the same group performed at T > 50°C with and without capsaicin also show that the addition of the vanilloid produces an increase in $P_o$ of at least twofold (*Yao et al., 2010*), suggesting that $P_o$ at high temperatures can be substantially less than 1. Macroscopic data from other laboratories also show less-than-maximal channel activation by temperature as compared with saturating capsaicin (*Cao et al., 2014*). One important consideration in comparing single-channel and macroscopic data is the fact that temperature-inactivated channels can still respond to saturating capsaicin (*Cao et al., 2014*), and that temperature-dependent channel inactivation (or rundown) also occurs in the presence of external $Na^+$, so that it is possible that a subset of channels in HEK293 cells cultured at 37°C are inactivated and therefore unresponsive to heat. This would lead to an underestimation of the $P_o$ in the absence of external $Na^+$ or in heat-activation experiments in the presence of $Na^+$ when measured from macroscopic current recordings, as in the present study. However, we think that this

underestimation is not large given our noise-analysis results in the absence of external Na$^+$ (*Figure 5—figure supplement 2D–G*) and the observation of frequent closures in our outside-out recordings where the activity of single channels can be discerned, indicating that the P$_o$ in the absence of external Na$^+$ is not maximal even at 30°C (*Figure 2A* and *Figure 4—figure supplement 1D*). Given that all P$_o$-T relations were scaled under similar conditions, a systematic underestimation of the P$_o$ would not change any of the conclusions of the present work.

## Non-stationary noise analysis for determining P$_o$

A hybrid non-stationary/stationary noise analysis method was used. The data were acquired at 100 kHz and filtered at 15 kHz, and pipettes were covered in Q-dope or dental wax to reduce noise and stray capacitances. Series resistance compensation (90% ) was used in all recordings. Whole-cell and pipette capacitances were also compensated. For experiments at T > 6°C, temperature was controlled using perfusion (*Figure 5—figure supplement 3A*). The temperature-controlled recording chamber (*Figure 4—figure supplement 1A*) was used for experiments at T < 6°C. The data were acquired at a single temperature per cell. For the WT channel, currents in response to 40–50 voltage pulses (3 s inter-pulse interval) of 400 ms duration from -90 to +90 mV were obtained at 0 Na$_o$, followed by another 50 pulses at 10 μM capsaicin + 130 Na$_o$. For TRPV1-Δ604–626, 10 traces were obtained in 0 Na$_o$ (not used for analysis), followed by 50 traces in 10 μM capsaicin + 130 Na$_o$, and finally 50 traces in 0 Na$_o$ + 10 μM capsaicin. For each experiment, the current variance ($\sigma^2_I$) was calculated from the ~50 current traces obtained with the same extracellular recording solution (i.e. 0 Na$_o$ or 130 Na$_o$ + caps, for the WT channel) as the ensemble average of the squared differences of successive currents records (*Heinemann and Conti, 1992*; *Alvarez et al., 2002*), an optimal method for channels with rundown. Briefly, we first calculated difference records y$_j$ between current traces x$_j$ elicited by successive voltage steps as: $y_j = \frac{1}{2}\left(x_j - x_{j+1}\right)$, where j goes from 1 to the total number N of current traces recorded at a given experimental condition (i.e. ~50 for most experiments). The variance $\sigma^2_I$ was then calculated from: $\sigma^2_I = \frac{2}{N-2}\sum_1^{N-1}(y_i - \overline{y})^2$, where $\overline{y}$ is the mean of the difference records. The mean current ($I_{mean}$) traces for each experiment were calculated by averaging together all ~50 current records obtained in the same experimental condition (see example in *Figure 5—figure supplement 2E*). To correct for leak and baseline variance, $\sigma^2_I$ and $I_{mean}$ calculated from the steady-state data points at -90 mV and 0 Na$_o$ (0 Na$_o$ + caps for TRPV1-Δ604–626) were subtracted from $\sigma^2_I$ and $I_{mean}$, respectively, measured at +90 mV at the other experimental condition(s). The initial and final points of the $\sigma^2_I$ and $I_{mean}$ traces at +90 mV were deleted due to contaminating capacitive currents that could not be fully compensated. Data for the two conditions (i.e. 0 Na$_o$ and 130 Na$_o$ + caps, for the WT channel) were then plotted together in $\sigma^2_I$ vs $I_{mean}$ plots (see example in *Figure 5—figure supplement 2D*), and fitted to *Equation 2*:

$$\sigma^2_I = I_{mean}i - \frac{I^2_{mean}}{N} \tag{2}$$

where *i* is the single-channel current amplitude and *N* is the number of channels in the cell. The open probability (P$_o$) at both conditions was then calculated from the two mean steady-state current ($I_{mean,ss}$) traces (such as those in *Figure 5—figure supplement 2E*) and the values for *i* and *N* obtained from fits of *Equation 2*, using *Equation 3*:

$$P_o = \frac{I_{mean,ss}}{iN} \tag{3}$$

(see *Figure 5—figure supplement 2G* for all calculated P$_o$-values for the WT channel at both 0 Na$_o$ and 130 Na$_o$ + caps.). The noise-analysis approach used here assumes that *i* and *N* are not changed by Na$^+$-removal or addition of capsaicin, since data obtained at two different conditions were plotted together in our variance vs mean plots, and fitted with the same parameters. We believe this assumption to be supported by several lines of evidence, as discussed above.

It is important to note that the single-channel current values at +90 mV estimated from noise analysis systematically underestimate *i* by a factor of ~2 when compared to single-channel recordings (*Figure 5—figure supplement 2F*). This underestimation was the same for experiments performed at all temperatures and also for both WT and Δ604–626 channels. When $\sigma^2_I$ vs $I_{mean}$ plots were fit by *Equation 2* while constraining the value of *i* based on single-channel recordings (see red fit in

*Figure 5—figure supplement 2D*), the resulting values for $N$ were considerably smaller than those obtained from unconstrained fits. However, the $P_o$ calculated using parameters from unconstrained fits are similar to that calculated using constrained fits (*Figure 5—figure supplement 2G*), indicating that the noise analysis provides approximate estimates of the $P_o$ that coincide with equivalent values obtained using single-channel recordings (*Premkumar et al., 2002*; *Hui et al., 2003*). Notably, the temperature-dependence of *i*-T relations obtained from the estimates of *i* using noise analysis is consistent with estimates we obtained from macroscopic I-T relations in saturating capsaicin and single-channel recordings (*Figure 5—figure supplement 2F*).

## Assumptions for the modeling of temperature-dependent gating of the TRPV1 channel

The allosteric model i is conceptually simple and contains the necessary features to qualitatively reproduce the most relevant aspects of our data pertaining to the effects of temperature, external $Na^+$ and capsaicin on the gating of TRPV1. A slightly simpler version of the model (with only a single temperature-dependent transition) has been used to accurately reproduce the effects of capsaicin, voltage and a narrow range of temperatures in TRPV1 (*Matta and Ahern, 2007*; *Yao et al., 2010*; *Gregorio-Teruel et al., 2014*; *Gregorio-Teruel et al., 2015*). However, four mechanistically relevant assumptions inherent to this model are not so evidently justified by the data, and require further discussion.

The most important assumption regarding the model is that the opening transition is temperature-independent. This feature of the model was incorporated in order to reproduce the plateaus observed in some $P_o$-T relations, in which the $P_o$ does not increase with temperature despite being well below its maximal theoretical value of 1.0 (*Figure 5E*). Are there other gating mechanisms that produce temperature-independent gating at $P_o < 1.0$ and have a temperature-dependent open/closed equilibrium? This is relevant given a recent study on Kv channels, which suggested that the opening transition of channels structurally related to the TRPV1 can be steeply temperature-dependent (*Yang et al., 2014*). Additionally, if the opening transition were responsible for the heat-activation of the TRPV1, a lot of information on the elusive mechanism of temperature-dependent gating could be readily gained from the structures of the channel in the closed and open states (*Cao et al., 2013*). A temperature-dependent open/closed equilibrium implies that the enthalpies ($H^\circ$) associated with the open and the closed states are different (i.e. $\Delta H^\circ$ is large). A modulator such as $Na^+$ could affect the temperature-dependence of channel opening by altering the enthalpy of the open or the closed state such that $\Delta H^\circ$ changes. In this scenario, a temperature-independent opening transition (with $\Delta H^\circ \approx 0$) could become temperature-dependent through the binding of sodium, if sodium increases the enthalpy of the open state or decreases the enthalpy of the closed state. The resulting $P_o$-T relations would be flat (no temperature-dependence) in the absence of $Na^+$, and steeply temperature-dependent in the presence of $Na^+$. However, our experimental results suggest that the channel retains temperature-dependence in the absence of external $Na^+$, as evidenced by the steep I-T relations at T > 30°C obtained from temperature jumps (*Figure 5D*) and the I-T relations at low temperatures in the presence of capsazepine without external $Na^+$ (*Figure 6A*). This indicates that external $Na^+$ is not directly responsible for conferring temperature-sensitivity to the TRPV1 channel. Thus, a model in which external $Na^+$ confers temperature-sensitivity to the opening transition would require distinct $Na^+$-independent heat-driven transitions to occur in the absence of external $Na^+$ in order to replicate our experimental results. Model ii (*Figure 7—figure supplement 1A*) incorporates these requirements and reproduces our experimental findings with comparable accuracy as model i (*Figure 7—figure supplement 1B*). However, model ii is mechanistically more complex than model i, since it requires distinct temperature-dependent transitions depending on whether the channel is bound to external $Na^+$ or not, implying that the TRPV1 possess different mechanisms of temperature-sensing. On the contrary, model i requires a single pair of temperature-dependent transitions and a common mechanism for temperature-sensing. Although a distinction between the two possibilities cannot be made, we favor model i due to its simplicity. Additionally, it can be concluded that the opening transition is not obligatorily temperature-dependent, but could become temperature-dependent under more complex models for TRPV1 channel gating.

The second important assumption is the inclusion of an additional temperature-dependent transition in model i. The inherent complexity of $P_o$-T relations in 130 $Na_o$ (*Figure 4C*) strongly suggest

that there are at least two temperature-dependent transitions involved in the gating mechanism of the TRPV1 channel. Furthermore, the $P_o$-T relations obtained with the turret-deletion construct in saturating capsaicin and external $Na^+$ unambiguously show the presence of at least two different temperature-dependent transitions associated with gating of TRPV1 (*Figure 9C* and *Figure 9—figure supplement 1B*). One transition appears to dominate temperature-dependent gating at T > 38°C, and represents the steep temperature-dependence that has been measured for TRPV1 in previous studies (*Liu et al., 2003*; *Voets et al., 2004*; *Yao et al., 2010*; *Cao et al., 2013*). Another transition dominates channel activity at lower temperatures, and appears to have a somewhat lower temperature-dependence, although this may only be the result of this temperature-dependent transition being close to saturation in the experimentally accessible range of temperatures. If the data is simulated using a model with a single temperature-dependent transition (Model iii, *Figure 7—figure supplement 1C*) with a large associated enthalpy, the resulting $P_o$-T curves at T < 35°C would be steeper than the experimental data. If the enthalpy for the temperature-sensor is chosen such that the model agrees well with the experimental data at T < 35°C, then the predicted $P_o$-T relations lack the highly temperature-dependent increase in $P_o$ at T > 38°C that is characteristic of the TRPV1 channel (see the predictions in *Figure 7—figure supplement 1D*), indicating that activation of the TRPV1 by heat involves at least two temperature-dependent transitions. Finally, we arranged the two temperature-dependent transitions in Model i sequentially, as opposed to considering them as separate coupled equilibria in a parallel arrangement. We chose a sequential arrangement of the transitions because it reduces the number of states in the model and eliminates a free parameter (the allosteric coupling constant between the two temperature-dependent transitions). However, fits to the experimental $P_o$-T relations of identical quality could be obtained with a parallel arrangement of temperature-sensors (data not shown), so that this aspect of the model is unconstrained.

A third assumption in Model i is that external $Na^+$ can bind to the TRPV1 in all of its possible conformations (i.e., there are no states to which sodium cannot bind). This has important implications for the mechanism of $Na^+$-regulation because one alternate mechanism by which sodium could inhibit the TRPV1 channel is by binding exclusively to the closed state. In that case, sufficiently high concentrations of external $Na^+$ should completely prevent opening of the channel by other stimuli. We can exclude this possibility because the channel can be robustly opened with capsaicin (*Figure 1C*) and heat (*Figure 4A*) in the presence of a saturating concentration of $Na^+$ (130 mM), or even when the concentration of extracellular $Na^+$ ions is increased to 600 mM (*Figure 7—figure supplement 2A*) or 2 M (data not shown). In addition, temperature-dependent activation of TRPV1 in the presence of 600 mM external $Na^+$ does not qualitatively differ from that measured with 130 mM external $Na^+$ (*Figure 7—figure supplement 2B*), suggesting that external $Na^+$ does not modulate the temperature-sensor by binding exclusively to its resting state.

The fourth assumption in the modeling was to neglect the effect of voltage-dependent gating on the calculated $P_o$-T relations, given that all experimental data were obtained at a fixed voltage. The low permeability of $NMDG^+$ relative to $Na^+$ greatly limited the range of voltages over which we could obtain useful data at most of the relevant experimental conditions in this study (see *Figure 1—figure supplement 4*). We therefore decided to estimate how much could changes in voltage above 0 mV alter the shape of $P_o$-T curves in both saturating capsaicin and in the absence of $Na^+$. To do this, we first calculated G-V relations in saturating capsaicin at two temperatures (22°C and 8°C) from families of currents elicited by voltage-steps from -120 to +140 mV (*Figure 1—figure supplement 3A* and *Figure 7—figure supplement 3A*). The G-V relations at both temperatures reach saturation at 0 mV (*Figure 7—figure supplement 3B*), indicating that voltage does not appreciably influence channel gating and its temperature-dependence in saturating capsaicin at positive membrane potentials, consistent with previous measurements (*Matta and Ahern, 2007*; *Yao et al., 2010*). Furthermore, this indicates that voltage-dependent changes in the I-V relations in saturating capsaicin at positive voltages reflect only the changes in single-channel current as a function of voltage but not changes in $P_o$. We therefore divided the I-V relations in the absence of external $Na^+$ (22°C, *Figure 1—figure supplement 3A*; 8°C, *Figure 7—figure supplement 3A*) by the I-V relations in saturating capsaicin to obtain the fraction of open channels ($P_o$) as a function of voltage for data at both temperatures (*Figure 7—figure supplement 3C*). The resulting $P_o$-V relations at the two temperatures superimpose, confirming that removal of external $Na^+$ leads to negligible temperature-dependence in channel gating over a range from 5 to 25°C. Moreover, the $P_o$-V relations show only weak voltage-dependence, suggesting that the effects of voltage-dependent gating on our $P_o$-T relations

measured at a fixed voltage should have a minimal influence on the interpretation of our results. We therefore decided not to include voltage-dependent transitions in our final model.

One final modification to the model was included to better account for the $Na^+$ and capsaicin concentration-dependence of $P_o$, since both modulators seem to exhibit weak cooperativity. Indeed, cooperative interactions between capsaicin and protons have been proposed to occur (*Ryu et al., 2003*), as well as positive cooperativity in the binding of other vanilloids (*Acs and Blumberg, 1994*). To explicitly introduce cooperativity would have required us to extend the model to take into account the tetrameric assembly of TRPV1, incrementing the number of states, free parameters and assumptions necessary to model the data. We therefore introduced an arbitrary cooperativity term for the binding of both $Na^+$ and capsaicin as is done in the Hill model of cooperativity, such that the modified capsaicin- or $Na^+$-dependent equilibrium constants had the form: $K_1 = (K_1' \times [Na])^{1.5}$ and $K_2 = (K_1' \times [caps.])^{1.5}$.

$P_o$ values from the models were calculated using programs written in Igor. Data was fit by visual inspection while manually varying the parameters.

## On the effect of heat capacity ($\Delta C_p$) on modeled $P_o$-T curves

The molecular mechanism underlying high-temperature sensitivity in thermo-TRP channels remains unknown. In this regard, it was recently noted that changes in side-chain hydration associated with a protein conformational transition that involves only a few residues could result in moderate changes in $\Delta C_p$, which in turn would render that conformational change highly temperature-sensitive (*Clapham and Miller, 2011*). The applicability of this molecular mechanism for ion channels was recently demonstrated using voltage-gated potassium channels with engineered voltage-sensing domains (*Chowdhury et al., 2014*). Given the possibility that $\Delta C_p$ is responsible for the temperature-sensitivity in TRPV1 channels, we tested whether considering a change in heat capacity for both temperature-dependent transitions in model i would have an influence on the behavior of the model and the interpretation of our results. Under this framework, equilibrium constants J(T) with the following form were selected (*Clapham and Miller, 2011*): $J(T) = e^{\left(\frac{\Delta S^o(T_0)}{R} - \frac{\Delta C_p}{R}\left(1 - \frac{T_0}{T} + \ln\left(\frac{T_0}{T}\right)\right)\right)}$, where $\Delta S^o(T_0)$ is the standard entropy change at reference temperature $T_0$, R is the gas constant and T the temperature (in Kelvins). Notably, the fits to the experimental data generated by model i with $\Delta C_p \neq 0$ are indistinguishable from those generated without considering a change in heat capacity ($\Delta C_p = 0$) (*Figure 7—figure supplement 4B*), as long as the respective equilibrium constants $J_1(T)$ and $J_2(T)$ take similar values in the experimentally explored range of temperatures (*Figure 7—figure supplement 4A*). No changes in parameters other than those associated with $J_1(T)$ and $J_2(T)$ were required (*Figure 7—source data 2E*). Thus, our results are equally compatible with temperature-sensing mechanisms that do or do not invoke changes in heat capacity.

## Simulating temperature-dependent inactivation

Comparison of I-T relations obtained using slow temperature ramps and fast temperature jumps in the absence of external $Na^+$ (*Figure 5D*) indicates that temperature-dependent inactivation can dominate the behavior of $P_o$-T relations, obscuring the high temperature-sensitivity of gating of the TRPV1 channel. This phenomenon is apparent for several of the experimental $P_o$-T curves measured in this study. However, since temperature-dependent inactivation is irreversible, it cannot be included in the mathematical models in this study, which are for equilibrium conditions. Nonetheless, we were interested in determining whether a simple decaying exponential function of time (to simulate inactivation) could effectively reduce the steepness of $P_o$-T curves and ultimately lead to a steep temperature-dependent decrease in $P_o$. The exponential function of time $f(t)$ had the form: $f(t, T) = e^{-k(T)t}$, where $t$ is time and $k(T)$ is a temperature-dependent rate constant given by: $k(T) = k_0 e^{-\frac{\Delta H^{\neq} - T\Delta S^{\neq}}{RT}}$, where $k_0$ is a pre-factor (see below), $\Delta H^{\neq}$ is the activation enthalpy for the inactivation process, $\Delta S^{\neq}$ is the activation entropy, $R$ is the gas constant and $T$ the temperature. The $P_o$-T relation with inactivation would then be given by: $P_{o,inact}(t, T) = P_o(T) \times f(t, T)$. The mechanism by which temperature-dependent inactivation depends on $Na^+$, open probability and temperature is unknown and beyond the scope of this study. However, we decided to incorporate $Na^+$-dependence in an arbitrary way that would yet allow us to qualitatively assess the effects of an exponentially decaying function of time and temperature on the $P_o$-T curves at several $Na^+$ concentrations. We

therefore arbitrarily assigned $k(T)$ a $Na^+$- and temperature-dependent pre-factor $k_0$ as follows: $k_0(T, [Na^+]) = A(1 - F_{Na})^2$, where A is a constant and $F_{Na}$ is the fraction of $Na^+$-bound states at equilibrium relative to the total number of states in model i. Finally, to include time in the final calculation of inactivating $P_o$-T curves we took advantage of the sigmoidal shape of our experimental temperature vs time plots (see the inserts on *Figure 4A* and *5A*), allowing us to have an empiric function to transform the temperature in the x-axis of $P_o$-T plots into time. The function was: $t = -0.83258 \times \ln\left(\frac{T_{max}}{T - 0.1} - 1\right) + 3.3181$, where $T_{max}$ is the maximal temperature (which was set to 333 K) and *T* is the temperature (in Kelvins). *Figure 7—figure supplement 5A* shows a calculated temperature vs time plot, which was then used to obtain $P_o$-T curves with inactivation (*Figure 7—figure supplement 5B*). This approach allowed us to replicate the overall shape of $P_o$-T curves with inactivation and their $Na^+$ dependence, confirming that temperature-dependent inactivation can indeed reduce the apparent steepness of $P_o$-T relations and ultimately leading to a steep decay in open probability. The parameters used in the modeling were: A = 200, $\Delta H^{\neq}$ = 45 kcal/mol, $\Delta S^{\neq}$ = 0.133 kcal/mol.

## Acknowledgements

We thank Mark Mayer, Joe Mindell, Shai Silberberg, Gilman Toombes and members of the Swartz lab for helpful discussions. We thank David Ide for help in manufacturing elements of the temperature-control system. This work was supported by the Intramural Research Program of the NINDS, NIH (to KJS) and by an NINDS Competitive Postdoctoral Fellowship (to AJO).

## Additional information

### Competing interests
KJS: Reviewing editor, *eLife.* The other authors declare that no competing interests exist.

### Funding

| Funder | Grant reference number | Author |
| --- | --- | --- |
| National Institute of Neurological Disorders and Stroke | NS002945 | Kenton J Swartz |

The funders had no role in study design, data collection and interpretation, or the decision to submit the work for publication.

### Author contributions
AJO, Conception and design, Acquisition of data, Analysis and interpretation of data, Drafting or revising the article; CB, Produced recombinant DkTx, Conception and design, Acquisition of data, Drafting or revising the article, Contributed unpublished essential data or reagents; KJS, Conception and design, Analysis and interpretation of data, Drafting or revising the article

### Author ORCIDs

Kenton J Swartz, http://orcid.org/0000-0003-3419-0765

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
