## [Decision Letter]

Thank you for submitting your work entitled "An external sodium ion binding site controls temperature-dependent gating in TRPV1 channels" for consideration by *eLife*. Your article has been favorably evaluated by Richard Aldrich (Senior editor) and three reviewers, one of whom, David Clapham, is a member of our Board of Reviewing Editors, and another is Baron Chanda.

The reviewers have discussed the reviews with one another and the Reviewing Editor has drafted this decision to help you prepare a revised submission.

Summary:

Jara-Oseguera, Bae, and Swartz carry out the most thorough biophysical investigation of the TRPV1 channel to date and establish the allosteric nature of capsaicin and temperature gating of this important ion channel. The general question is how TRPV1 responds to small temperature changes. The results demonstrate that the TRPV1 channel has an external Na binding site that must be occupied for the channel to remain closed at physiological temperature, and is also likely responsible for acid sensing. The Na^+^ regulatory mechanism thus tunes TRPV1 for noxious stimuli. The authors find evidence for multiple temperature-sensitive transitions that couple to a relatively temperature-insensitive opening transition; the first temperature-sensitive transition maintains the channel in a closed state at physiological temperatures, whereas the second transition functions to activate the channel in response to noxious heat. Na^+^ inhibition has only very weak voltage-dependence, and there is substantial evidence in this paper that the site is near the turret, likely involving E600 in the extracellular loop just after the S5 helix and close to the DkTx binding site. This is a solid contribution to the field and for the first time provides evidence for multiple temperature-dependent transitions in the TRPV1 channel. Nevertheless, we have some concerns mainly with regards to the interpretation of their data and its implications.

Points to consider in the revision:

1) The data and their models are consistent with an allosteric site for sodium binding which when occupied favors channel closure. This allosteric site like all the other allosteric sites has an independent equilibrium constant and is coupled to pore opening with a coupling constant. It also coupled to temperature-sensors and capsaicin binding site with coupling constants of different strengths. However, the manuscript is presented as if the sodium binding site is uniquely involved in temperature-dependent gating. The title states that "sodium ion binding site controls temperature-dependent gating" whereas, in our opinion, it would be more appropriate to state that "sodium ion binding site is allosteric modulator of pore gating". The sodium ion binding site is as much of an influence on temperature-dependent gating as capsaicin, protons or voltage-sensor which incidentally was not included in this model. While the authors might argue that the coupling constants that link sodium ion binding site to the temperature-sensor activation is much higher than those linking capsaicin binding site, we think that simply reflects the problem of parameter identifiability. It is unclear whether these equilibrium parameters are unique especially in absence of kinetic constraints. For instance, if these model parameters are correct, then the coupling constants that link temperature dependent transition to pore opening is much smaller than those that link capsaicin binding or sodium activation to channel opening (models i and iii). Therefore, by this logic the TRPV1 channels are only weakly activated by temperature. In light of these uncertainties, it is incumbent on the authors to interpret their models with a degree of parsimony. These caveats should be also reflected in their Abstract.

2) The calculated P_o_ in presence of full sodium is much smaller than 0.1 even at their highest temperatures (Figure 4). Previous work from Feng Qin's lab using single channel measurements (Biophysical Journal 2003) showed that the Po values for TRPV1 channels at 50 degrees reach very close to 1. At 45 degrees, their P_o_ values were 0.5 whereas in this manuscript the Po values are approximately 0.06 at those temperatures. Why is there so much discrepancy in the Po values even though both groups are studying the same channel expressed in Hek cells? Qin and coworkers obtained their P_o_ measurements directly from single channel recordings whereas in this manuscript P_o_ is deduced from macroscopic currents after correcting for temperature-dependent changes in conductance.

3) The authors do not have strong evidence to suggest that the temperature-sensor is allosterically coupled to the pore. A hallmark of allosteric system is that one should be able to observe temperature-independent transitions at both high and low temperatures. According to their models, the saturation at low temperatures will occur at ranges that would be experimentally intractable and at high temperatures this would be complicated by inactivation (or desensitization?). This is an unfortunate situation that ultimately makes it difficult to discriminate between an allosteric versus an obligatorily coupled temperature-sensor. To be fair, the authors acknowledge this problem and state that they favor the allosteric model because it is simpler. I would argue that an obligatorily coupled model has fewer free parameters than an allosteric model. Again these caveats should be clearly stated in their conclusion.

4) TRPM7 is ubiquitous and contributes to outward currents in all cells we have examined. This current runs up over time if Mg is omitted from the pipette (free Mg < 1 mM). The authors have used 10 mM Mg inside the cell in temperature experiments, but they might want to check in other cases where capsaicin, DkTx, and other experimental manipulations are carried out without temperature changes that the currents are not contaminated by M7.

---

## [Author Response]

We thank the reviewers for so carefully reading and thinking about our manuscript, and for raising a number of very germane issues. We have now addressed all of their concerns, as described in detail below. In addition to the changes suggested by the reviewers, we have made a couple of additional modifications to the manuscript which we will describe first:

1) For all figures depicting structures (Figure 1, Figure 3—figure supplement 1 and Figure 10—figure supplement 1), we have now used our refined structural models of the TRPV1 in the apo and the DkTx/RTx bound states, which includes the docked solution structure of the DkTx tarantula toxin. These models are described in another publication currently in press in *eLife* (Bae et al., 2016), and have been refined to contain significantly less structural violations and contain energy minimized positions for side chains and flexible loops. We also make reference to a possible mechanism for activation of the TRPV1 channel by DkTx inferred from that study, which points to a site on the extracellular pore which is affected by the binding of the toxin and that could potentially be involved in temperature-sensing.

2) We have included data showing that extracellular magnesium can directly activate the TRPV1 (Figure 3—figure supplement 1). This is an important point, as two recent studies in which external sodium was replaced with magnesium proposed that the divalent ions affect the temperature-sensitivity of the channel without taking into consideration the inhibitory effect of sodium on TRPV1 (Cao et al., JGP, 2014 and Yang et al., JGP, 2014).

Points to consider in the revision:

*1) The data and their models are consistent with an allosteric site for sodium binding which when occupied favors channel closure. This allosteric site like all the other allosteric sites has an independent equilibrium constant and is coupled to pore opening with a coupling constant. It also coupled to temperature-sensors and capsaicin binding site with coupling constants of different strengths. However, the manuscript is presented as if the sodium binding site is uniquely involved in temperature-dependent gating. The title states that "sodium ion binding site controls temperature-dependent gating" whereas, in our opinion, it would be more appropriate to state that "sodium ion binding site is allosteric modulator of pore gating". The sodium ion binding site is as much of an influence on temperature-dependent gating as capsaicin, protons or voltage-sensor which incidentally was not included in this model. While the authors might argue that the coupling constants that link sodium ion binding site to the temperature-sensor activation is much higher than those linking capsaicin binding site, we think that simply reflects the problem of parameter identifiability. It is unclear whether these equilibrium parameters are unique especially in absence of kinetic constraints. For instance, if these model parameters are correct, then the coupling constants that link temperature dependent transition to pore opening is much smaller than those that link capsaicin binding or sodium activation to channel opening (models i and iii). Therefore, by this logic the TRPV1 channels are only weakly activated by temperature. In light of these uncertainties, it is incumbent on the authors to interpret their models with a degree of parsimony. These caveats should be also reflected in their Abstract.*

We have now modified the title of the manuscript, the Abstract and the main text to emphasize a more general role of the external sodium site in the allosteric gating of TRPV1. We also now give more emphasis to the effects of capsaicin on temperature-sensitivity and gating in general. A small restructuring in the order of the figures in the paper has allowed us to provide a more compelling and precise discussion on the role of capsaicin on temperature-sensitivity. While the large number of free parameters in the model and the small number of experimental constraints don’t allow us to make comparisons between the relative magnitudes of the coupling constants for external sodium and capsaicin, we believe that the observation that P_o_-T relations at all subsaturating concentrations of capsaicin exhibit temperature dependence at low temperatures, in contrast to what we observe in the absence of external sodium, suggests that capsaicin has a much larger influence on the open/closed equilibrium than on the function of the temperature-sensor. Otherwise, no temperature-dependent changes in P_o_ at low temperatures would be observed in the presence of capsaicin, as is observed in the case of DkTx. This is now discussed more clearly in the text. We have also incorporated a discussion on the possible alternatives to the model proposed here in the main text to stress that other alternative models are theoretically possible, but while also emphasizing the qualitative features of the data that are independent of the model. Finally, we now give equal weight to the allosteric effect of external sodium on temperature-dependent gating and on the open/closed equilibrium, as there is no way of drawing any quantitative conclusions on the relative magnitudes of the strength of the couplings between external sodium and the two other equilibria.

2) The calculated P_o_ in presence of full sodium is much smaller than 0.1 even at their highest temperatures (Figure 4). Previous work from Feng Qin's lab using single channel measurements (Biophysical Journal 2003) showed that the Po values for TRPV1 channels at 50 degrees reach very close to 1. At 45 degrees, their P_o_ values were 0.5 whereas in this manuscript the Po values are approximately 0.06 at those temperatures. Why is there so much discrepancy in the Po values even though both groups are studying the same channel expressed in Hek cells? Qin and coworkers obtained their P_o_ measurements directly from single channel recordings whereas in this manuscript P_o_ is deduced from macroscopic currents after correcting for temperature-dependent changes in conductance.

It is true that the study mentioned reports high P_o_ values for heat activation of TRPV1, whereas our study as well as several others using macroscopic measurements show that heat is not as effective as saturating capsaicin at opening the channel. We think that one source of the difference is that there may be a fraction of channels that are heat-inactivated in macroscopic recordings, leading to an underestimation of P_o_. However, we think that there is variability in the maximal P_o_ for heat activation, because our noise analysis is roughly consistent with our method for calculating P_o_ from macroscopic currents and our recordings where unitary channel activity can be discerned are not consistent with a P_o_ near 1. We have added a discussion of this issue in the Methods section (subsection “Scaling of Po-T relations based on macroscopic current recordings“, last paragraph), and pointed out how this does not affect any of the conclusions presented in our study.

*3) The authors do not have strong evidence to suggest that the temperature-sensor is allosterically coupled to the pore. A hallmark of allosteric system is that one should be able to observe temperature-independent transitions at both high and low temperatures. According to their models, the saturation at low temperatures will occur at ranges that would be experimentally intractable and at high temperatures this would be complicated by inactivation (or desensitization?). This is an unfortunate situation that ultimately makes it difficult to discriminate between an allosteric versus an obligatorily coupled temperature-sensor. To be fair, the authors acknowledge this problem and state that they favor the allosteric model because it is simpler. I would argue that an obligatorily coupled model has fewer free parameters than an allosteric model. Again these caveats should be clearly stated in their conclusion.*

We have now removed all statements in the Abstract pertaining to the temperature-sensitivity of channel opening, and included a more extensive discussion on this subject in the main text. Two different issues are now mentioned: 1) whether the opening transition is itself temperature-dependent and 2) whether the channel requires activation of the temperature sensor in order to activate. We consider that the observation of temperature-independent plateaus in P_o_-T relations at P_o_ values lower than 1 suggests that the opening transition itself is not obligatorily temperature-dependent, as the channel can still be closed by voltage while on the plateaus. This does not preclude the possibility of the activation gate of TRPV1 transitioning to various conformations, with some of these transitions occurring in a temperature-dependent manner. However, we consider this possibility to be mechanistically more complex than the model proposed in the manuscript, which is now described explicitly in the main text. Regarding the need of the temperature-sensor to activate in order for the channel to open, we now mention in the main text that this remains an open question, but that we have favored the more general mechanism that considers all possible transitions, including opening of channels with temperature sensors in the resting state. We have also modified the pictorial depiction of our model, which aids in the treatment of these issues.

*4) TRPM7 is ubiquitous and contributes to outward currents in all cells we have examined. This current runs up over time if Mg is omitted from the pipette (free Mg < 1 mM). The authors have used 10 mM Mg inside the cell in temperature experiments, but they might want to check in other cases where capsaicin, DkTx, and other experimental manipulations are carried out without temperature changes that the currents are not contaminated by M7.*

Thank you for pointing out the identity of the channel that we have identified as contaminating our results in some of our recordings from HEK cells. Typically our whole-cell recordings have very small leak currents at negative voltages, with some exceptions that become more frequent as the passage number of the culture increases. TRPV1 displays prominent outward rectification, and in our recording conditions, the contaminating endogenous currents would become evident as larger inward currents at negative voltages and a reduction of the outward-rectifying character of IV relations for TRPV1. We systematically discarded cells that displayed less prominent outward rectification. We also noticed that these endogenous currents become more prominent at elevated temperatures (since they start to increase at T > 20 °C), which is the reason why we included magnesium in the pipette solution for all heat-activation experiments. We had originally performed quite a few experiments at high-temperature without magnesium, which we repeated once we realized that these currents sometimes contaminated our recordings. We also checked that adding magnesium does not influence IV relations in the absence of external Na^+^ or after addition of DkTx, and all of our experiments at room temperature display pronounced outward-rectification, so we are confident that these reflect the activity of TRPV1. We have now detailed this in the Methods section.